# Impact of Compound Flood Event on Coastal Critical Infrastructures Considering Current and Future Climate

Mariam Khanam[1], Giulia Sofia[1], Marika Koukoula[1], Rehenuma Lazin[1], Efthymios I. Nikolopoulos[2], Xinyi Shen[1], and Emmanouil N. Anagnostou[1]

[1]Civil and Environmental Engineering, University of Connecticut, Storrs, CT 06269, USA
[2]Mechanical and Civil Engineering, Florida Institute of Technology, Melbourne, FL 32901, USA

*Correspondence to*: Anagnostou, Emmanouil N. (emmanouil.anagnostou@uconn.edu)

**Abstract.** The changing climate and anthropogenic activities raise the likelihood of damages due to compound flood hazards, triggered by the combined occurrence of extreme precipitation and storm surge during high tides, and exacerbated by sea level rise (SLR). Risk estimates associated with these extreme event scenarios are expected to be significantly higher than estimates derived from a standard evaluation of individual hazards. In this study, we present case studies of compound flood hazards affecting critical infrastructure (CI) in coastal Connecticut (USA) based on actual and synthetic (considering future climate conditions for the atmospheric forcing, sea level rise, and synthetic hurricane tracks) hurricane events, represented by heavy precipitation and surge combined with tides and SLR conditions. We used the Hydrologic Engineering Center's River Analysis System (HEC-RAS), a two-dimensional hydrodynamic model to simulate the combined coastal and riverine flooding on selected CI sites. We forced a distributed hydrological model (CREST-SVAS) with weather analysis data from the Weather Research and Forecasting (WRF) model for the synthetic events and from the National Land Data Assimilation System (NLDAS) for the actual events, to derive the upstream boundary condition (flood wave) of HEC-RAS. We extracted coastal tide and surge time series for each event from the National Oceanic and Atmospheric Administration (NOAA) to use as the downstream boundary condition of HEC-RAS. The significant outcome of this study represents the evaluation of changes in flood risk for the CI sites for the various compound scenarios (under current and future climate conditions). This approach offers an estimate of the potential impact of compound hazards relative to the 100-year flood maps produced by the Federal Emergency Management Agency (FEMA), which is vital to developing mitigation strategies. In a broader sense, this study provides a framework for assessing risk factors of our modern infrastructure located in vulnerable coastal areas throughout the world.

## 1 Introduction

The impacts of hurricanes such as Harvey, Irma, Sandy, Florence, and Laura are characteristic examples of hazardous storms that have affected the society and environment of coastal areas, and have damaged infrastructure, through the combination of heavy rain and storm surge. The increased frequency of such events raise concerns about compound flood hazards previously considered independent of one another (Barnard et al., 2019; Leonard et al., 2014; Moftakhari et al., 2017; Wahl et al., 2015; Zscheischler et al., 2018; Winsemius et al., 2013; Hallegatte et al., 2013; de Bruijn et al., 2017; de Bruijn et al., 2019, Bevacqua et al., 2019).

Concurrent with the rise in event intensities, the elevated damage and disruption caused by compound flooding (CF)
to critical infrastructure (CI) and services, including electrical systems, water, and sewage treatment facilities, and
other utilities that underpin modern society, have substantial adverse socioeconomic impacts, especially in low-lying
coastal areas, where almost 40 percent of people in the United States live (NOAA, 2013).
The growing record of significant impacts from extreme events around the world (Chang et al., 2007; McEvoy et al.,
2012; Ziervogel et al., 2014; FEMA, 2013; Karagiannis et al., 2017) adds an urgency to the need for reassessing CI
management policies based on compound impact, to help ensure flood safety and rapid emergency management
(Pearson et al., 2018).   The uncertainty of the current evolution of compound events translates into an even greater
uncertainty concerning future damage to CI (de Bruijn et al., 2019, Marsooli et al., 2019).
Recent studies have underlined the importance of understanding and quantifying the flood impacts on critical
infrastructure, and their broader implications in risk management and catchment-level planning (Chang et al., 2007;
McEvoy et al., 2012; Ziervogel et al., 2014; de Bruijn et al., 2019; Pearson et al., 2018; Pant et al., 2018; Dawson,
2018). Some authors have estimated the frequency of compound flooding and provide approaches to risk assessment
based on the joint probability of precipitation and surge (Bevacqua et al., 2019; Wahl et al., 2015). The spatial extent
and depth of compound flooding can vary in frequency (Quinn, et al., 2019) if any of the component of CF is not taken
into consideration while evaluating flood frequency. Both storm surges and heavy precipitation, and their interplay,
are likely to change in the future (Field et al., 2012, Dottori et al., 2018; Blöschl et al., 2017; Muis et al., 2016; Marsooli
et al., 2019; Vousdoukas et al., 2018). Nonetheless, the effects of CF, considering the climate change impact, have not
been thoroughly explored yet.

To deal with CF threats and challenges to coastal communities, there is a need to develop efficient frameworks for
performing systematic risk analysis based on a wide range of actual and what-if scenarios of such events in current
and future climate conditions. In this study, we focused on coastal power grid substations as critical infrastructure and
investigated the impacts of compound flood hazard scenarios associated with tropical storms. We present a hydrologic-
hydrodynamic modeling framework to evaluate the integrated impact of flood drivers causing CF by synthesising
current and future scenarios. This study enables the quantitative measurement of CF hazard casted on critical
infrastructures in terms of flood depth and flood extent by observing actual storm-induced floods and drawing
information from synthetic scenarios. To project the combined flood hazard in future climate conditions, we integrated
the effects of SLR, tides, and synthetic hurricane event simulations into the flood hazard exposure.
Even though past research on the assessment of damages to the power system components or other related
infrastructures has proposed design and operation countermeasures and remedies (i.e. Kwasinski et al. 2009; Reed et
al. 2010; Abi-Sarma and Henry, 2011; Chang et al., 2007; de Bruijn et al., 2019; Pearson et al., 2018; Pant et al., 2018;
Dawson, 2018), these studies lack a comprehensive hazard assessment on power grid components, and.potential
changes due to climate change.
The scenario-based analysis of this study formed the basis on which to address two questions:
*(1) What are the characteristics of the tropical storm-related inundation, considering the compound effect of riverine*
*and coastal flooding coinciding or not with peak high tides*
*(2) Will future climate (including SLR and intensification of storms due to warmer sea surface temperatures) bring a*
*significant increase in flood impact for the power-grid coastal infrastructures?*
The proposed framework offers a multi-dimensional strategy to quantify the potential impacts of tropical storms, thus
enabling for a more resilient grid for climate change and the increasing incidence of severe weather.
We investigated these questions based on eight case studies of CI in Connecticut (USA), distributed on the banks of
coastal rivers discharging along the Long Island Sound.

## 2 Materials and methods

### 2.1 Study sites

This study focused on seven coastal river reaches (Fig. 1, Table 1), where eight power grid substations lie in proximity
to riverbanks and are prone to flooding caused by both coastal storms (such as hurricanes) that combine heavy
precipitation and high surge. These power grid substations are coded on the map CI1 through CI8.
For each river reach adjacent to a CI, we developed a hydrodynamic model domain, and we applied a distributed
hydrological model for predicting river flows from the upstream river basin. Table 1 shows the specification of each
river reach, associated drainage basin, the correspondent domain extent for the hydrodynamic simulations, and the
hydrological distance [distance along the flow paths] of each power grid substation from the coastline. This distance
was derived using the 30m National Elevation Dataset (NED) for the continental United States (USGS 2017).
Among the case study sites, two CIs are relatively inland [CI3 and CI4] (table 1: see hydrologic distance. Figure 1:
see coastal boundary), nonetheless all the sites are included within the Coastal Area as defined by Connecticut General
Statute (CGS) 22a-94(a) [https://www.cga.ct.gov/current/pub/chap_444.htm#sec_22a-94]. The considered rivers
belong to watersheds ranging from 10 to 300 $km^2$ basin area, which are sub-basins of the Connecticut River basin.
The hydrodynamic model simulation domains ranged from 3.7 to 8.3 km in river length and 2.2 and 20.7 $km^2$ in area.

### 2.2 Simulation framework

To evaluate the effect of compound events, we selected four tropical storms: two actual hurricanes (Sandy and Irene)
that hit Connecticut, and two synthetic scenarios based on actual hurricanes Sandy and Florence. Both Irene (August
21–28, 2011) and Sandy (October 22–November 2, 2012) reached category 3, but they made landfall in Connecticut
as category 1 hurricanes. The synthetic simulations (Chapt. 2.2.1) include different atmospheric conditions leading to
landfall scenarios with greater impacts. The Sandy synthetic scenario represents hurricane Sandy under future climate
and sea surface conditions (Lackmann 2015), while the synthetic scenarios for Florence were based on simulated
surge-tide condition and future SLR (see Chapt. 2.2.1 and 2.3).
To investigate the impact of floods of the various scenarios, we devised a combined hydrological (Chapt. 2.2.2) and
hydrodynamic (Chapt. 2.2.3) modeling framework (Figure 2), forced with weather reanalysis and geospatial data for
the actual events, and a numerical weather prediction model (subsection a) for the synthetic events (that is, synthetic
hurricane Florence and future hurricane Sandy).

### 2.2.1 Atmospheric simulations

To simulate the two synthetic Sandy and Florence hurricane events, we used the Weather Research and Forecasting (WRF) system (Powers et al., 2017; Shamarock et al., 2007). For the synthetic hurricane Florence event, we used a hurricane track forecast by the National Oceanic and Atmospheric Administration (NOAA), that as of September 6, 2018, according to the Global Forecast System (GFS) forecasts of the National Center for Environmental Prediction (NCEP), showed landfall in Long Island and Connecticut on September 14 as a category 1 hurricane (Higgins 2000). We based synthetic hurricane Sandy event on future climate conditions (post-2100).

For the soil type and texture input in the WRF model for both synthetic storm simulations, we used the USGS GMTED2010 30-arc-second (Danielson and Gesch 2011) Digital Elevation Model for the topography, the Noah-modified 21-category IGBP-MODIS (Friedl et al., 2010) for land use, and vegetation input, and the Hybrid STATSGO/FAO (30-second) (FAO 1991) for soil characteristics.

To simulate the synthetic hurricane Florence with WRF, we used the GFS forecasts at 0.25° x 0.25° spatial resolution as initial and boundary conditions. We used a three-grid setup with a coarse external domain of 18 km spatial resolution and two nested domains with 6 km and 2 km horizontal grid spacing, respectively. Two-way nesting was activated for both inner domains. Vertically, the domains stretched up to 50 mb with 28 layers. We parameterized convective activity on the outer (resolution of 18 km) and the first nested (resolution of 6 km) domain using the Grell 3D ensemble scheme (Grell and Devenyi 2002). Further details on the model setup are presented in Table 2.

For the future hurricane Sandy scenario, we used the hurricane Sandy simulations under future climate conditions (after 2100) by Lackman (2015), who used a three-grid setup at spatial resolutions of 54, 18, and 6 km. We defined initial and boundary conditions by altering the European Centre for Medium-Range Weather Forecasts (ECMWF) interim reanalysis (Dee et al., 2011) data, based on five General Circulation Model (GCM)-projected, late-century thermodynamic changes derived from the IPCC (Intergovernmental Panel on Climate Change) AR4 A2 emissions scenario (Meehl et al., 2017). A complete description of the modeling framework is provided by Lackman (2015).

### 2.2.2 Hydrological modeling

To account for the river inflow (upstream boundary condition), we applied a physically-based distributed hydrological model [CREST-SVAS (Coupled Routing and Excess Storage–Soil–Vegetation–Atmosphere–Snow)] described in Shen and Anagnostou (2017).

To simulate river discharges for the synthetic hurricanes (Florence and future Sandy), we used the WRF simulations at 6-km/hourly spatiotemporal resolution, as described above. To force the hydrological model for the actual events (Sandy and Irene), we used data from Phase 2 of the North American Land Data Assimilation System (NLDAS-2) (Xia et al., 2012) dataset. NLDAS-2 is a gridded dataset derived from bias-corrected reanalysis and in situ observation data, with a one-eighth-degree grid resolution and an hourly temporal resolution, available from January 1, 1979, to the present day. We derived the precipitation from daily rain gauge data over the continental United States, and all other forcing data came from the North American Regional Reanalysis (NARR) by NCEP (Higgins 2000), to which we applied bias and vertical corrections. To reduce the computational effort, we performed the hydrological simulation using a hydrologically conditioned 30 m spatial resolution DEM (USGS 2017).

The hydrologic simulation include the use of land use and land cover information retrieved from the Moderate
Resolution Imaging Spectroradiometer ("MOD12Q1" from MODIS) (Friedl et al., 2015). To compensate for the
coarse resolution (500 m) of these data, we obtained imperviousness ratios using Connecticut's Changing Landscape
(CCL) database and the National Land Cover Database (NLCD) at 30 m resolution. In CREST-SVAS, the land surface
process was simulated by solving the coupled water and energy balances to generate streamflow at hourly time steps
at the outlet of the studied watershed. CREST-SVAS was calibrated and validated for the whole Connecticut river
basin [that contains all the investigated sites] with an NSCE of 0.63 (Shen and Anagnostou, 2017). We further
validated the model considering hourly flows in two locations within the Housatonic River and Naugatuck River
watersheds with an NSCE of 0.69 (Hardesty et al., 2018). The quality measures indicate a satisfactory model
performance at the watershed scale over the topographic region that collectively include our study sites.
**2.2.3 Hydrodynamic modeling**
To assess the flood hazard in terms of extent and the maximum depth of the flood, we implemented the Hydrologic
Engineering Center's River Analysis System (HEC-RAS), developing two-dimensional model domains around the CI
location. Except for CI4 and CI5, which are within the same simulation domain, each substation has an independent
domain.
The inundation maps are derived using a 1m LIDAR DEM (CtECO 2016) taken as base maps for the study reaches.
To better represent the impacts of urban establishments on inundation dynamics, solid urban features such as houses
and buildings, which obstruct the flow of stormwater, were added to the bare-earth DEM. For this, we considered the
building footprints from (CtECO, 2012) and identified positions of buildings and houses in the DEM by increasing
the elevation of the pixels within the building footprint polygons by an arbitrary height of 4.5 m, assuming one-story
buildings.
The considered locations have no bathymetric (underwater topography) data represented in the DEM. In general, the
impact of inclusion/exclusion of bathymetry data on the hydrodynamic model simulations will vary according to the
river size and event severity (Cook & Merwade 2009). For the investigated events in this study flood risk is mainly
dominated by defence overflow and defence breaching. This means that we do not require detailed bathymetric
information in the upstream main channel, thereby considerably simplifying the modeling problem (Bates et al. 2013).
Therefore, we did not represent the flow of water in the main channel. Rather boundary conditions were given as time
series of water surface elevation imposed along the defence crests.
To reduce the computation time, we created a 2D mesh grid at 10 m background resolution, enforced with breaklines
to intensify the riverbank and other areas with a large elevation gradient up to 1 m resolution. The upstream boundary
condition was provided by CREST-SVAS, and the downstream boundary condition (coastal water level, including
coastal tide, storm surge, and sea level) was derived from National Water Level Observation Network (NWLON)
data, provided by NOAA. These data are available as actual observations and predictions at intervals of six minutes
to one hour. Figure 3 provides an example of one of the sites, showing the upstream and downstream boundaries,
along with a map overlay of flooded areas of five (SD1–SD5) scenarios (see below) for CI2. We initiated the
simulation with a warmup period of 12 hours to achieve stability. We chose the full momentum scheme in HEC-RAS
and extracted hourly output from the simulation.
The model parameters were calibrated to obtain realistic water depths and extents, as compared to reference data
collected for Sandy. To validate the hydrodynamic model simulations, we used surveyed HWMs (high water marks)
(Koenig et al., 2016) collected by the United States Geological Survey (USGS) after hurricane Sandy at 15 selected
locations spread across the simulation domains.  HWMs are frequently used to calibrate and validate model outputs
and satellite-based observations of flood depth (Bunya et al. ,2010; Cañizares and Irish 2008; Cariolet, 2010; Chang
et al., 2007; Hostache et al. 2009; McEvoy et al., 2012; Pearson et al., 2018;  Schumann et al., 2008; Schumann et al.,
2007; Schumann et al., 2007; Ziervogel et al., 2014). As for the flood extent, we further validated the model against
the most accurate available information on the 2D extent and maximum depth of storm surge for Sandy (FEMA, CT
DEEP, 2013), created from field-verified HWMs and Storm Surge Sensor data from the USGS.
An HWM does not necessarily indicate the maximum flood depth; rather, it can be a mark from a lower depth that
lasts long enough to leave a trail. Based on this understanding, we compared the HWMs against the simulated flood
depths within a 10x10m radius around the high water marks, also to avoid issues due to the presence of buildings in
the DEM (Boxplots in Fig. 4). The simulated depths demonstrated reasonable agreement with the collected HWM
values (Figure 4), with the model showing a slight overestimation. In this case, the systematic error fell within values
of expected precision, implying a consistent positive bias in the simulations not strong enough to hinder the results.
Figure 5 shows a visual comparison for CI1 and CI2 between the simulated inundation (Fig.5 a, c), and the reference
extent (Fig. 5 d,e). A slight overestimation of the flood level, ranging between 0.2 and 0.4 m, with a precision of 0.2
m or less, is observed for the inundation depths at the displayed locations, which is consistent with the results obtained
locally, at the HWM locations (Fig. 4). Taking into consideration the accuracy of the inundation depth, the declared
DEM accuracy (vertical RMSE ~0.3m), and the simplified modeling problem concerning bathymetry, the accuracy of
the flood extent assessment was judged satisfactory.

## 2.3 Compound scenarios

We modeled four types of synthetic compound event scenarios, as well as actual events by (1) simulating the synthetic
hurricanes; (2) introducing a climate change factor, in the form of SLR (~0.6 m), as projected for 2050, as a prediction
for intermediate low probability (CIRCA 2017); (3) shifting the surge timing to make the surge peak-level occurring
at local high tide; and (4) combining the SLR with the high tide condition. The combination of these four event types
yielded nine simulations, hereby coded as IR or SD for hurricanes Irene and Sandy, and FL for the synthetic hurricane
Florence.
Two scenarios were created for hurricane Irene. IR1 was the actual hurricane Irene, that made landfall in Connecticut
during high tide, and IR2 was the IR1 scenario with future SLR added to the tidal water level as a downstream
boundary condition in HEC-RAS.
For hurricane Sandy, we generated five scenarios. SD1 was the actual Sandy. For SD2, we shifted the peak high tide
to coincide with the maximum storm surge recorded, as derived from the local NOAA stations (hereafter referred to
'shifted tide water levels'). We further added SLR to the shifted tide water levels from SD2 to create the third scenario
(SD3). The remaining two scenarios for hurricane Sandy represented future climate conditions. Specifically, SD4 was
the future hurricane scenario simulated with the GFS (Chapt. 2.2.1) and shifted tidal water level. SD5 was the future
Sandy with shifted tide water levels and SLR.
For the synthetic hurricane Florence event, we simulated two scenarios. FL1 was the synthetic Florence event, based
on the GFS track that gave landfall in Connecticut and Long Island (Chapt. 2.2.1). FL2 was the same synthetic event,
with SLR added to the coastal water levels.
Table 3 shows, for each scenario, the basin-averaged event accumulated precipitation (mm) and the simulated peak
flow (m3/s) used as an upstream boundary condition in HEC-RAS, along with the recurrence interval of the peak
flows derived using a Log-Pearson probability distribution fitted using yearly maxima from the long-term simulated
flows (1979-2019) from CREST. This shows how significant the precipitation forcing was for each considered
scenario. For CI1, for example, the future Sandy (SD4/5) scenario, with a peak flow of 242.4 m3/s, was the most
extreme event with a recurrence interval of 316 years, followed by Irene (158.5 m3/s) and Florence (51.3m3/s) with
a recurrence interval of 56 and 2 years respectively, whereas, for CI8, Florence and future Sandy had similar
magnitudes with peak flows of 93.1m3/s (6) and 94.7m3/s (6), respectively. In table 3, we have summarised the
maximum total water level (tide & surge) used in the model at the downstream of the study sites for all the scenarios.
This table represents the change in the severity of the coastal component of the compound scenarios concerning added
challenges like shifted tide and SLR. For example, for CI3, the total water level increases 1m with the shifted tide
(SD2/ SD4), and with SLR it becomes 4.4 m.

## 2.4 Compound flood hazard analysis

We investigated the compound effect of the different events by comparing flood area extents and flood depths for
each event. For the flood area extent, we used as a baseline the 100-year flood maps provided by FEMA. The distance
correlation index (dCorr) (Székely et al; 2007) has been used to identify the correlation of the differences between
simulated and FEMA extent and compound events' parameters [flow and total water level peak]. dCorr values range
from 0 to 1 expressing the dependence between two independent variables. The closer the value to 1 is the stronger
the dependency would be, and zero implies that the two variables in question are statistically independent. dCorr can
depict the non-monotonic associations of the variables and declare the dCorr value is zero if only the variables are
statistically independent.
For the flood level differences, we considered the overall distribution of water depths across the domain of the CI sites
and investigated the time series of water depth at each location (Figure 6 is an example of the simulated flood depth
during the scenarios of Sandy (SD1- SD5) over time for CI2).
To evaluate the flood hazard in terms of flood depth, we computed a Cumulative Distribution Function (CDF) to
shows the probability that the flood depth will attain a value less than or equal to each measured value. We estimated
the CDF using all the depth values of all the grid of simulation domain, for the time step when the inundation was
maximum. We evaluated the depth empirical exceedance probability (Hanman et al., 2016; Lin et al., 2016; Warner
and Tissot 2012) within the whole domain, considering the maximum depth at each pixel, as suggested in (Pasquier
et al. 2019, Hamman et al. 2016). The benefits of this empirical approach are that it overcomes sensitivity to the
choice of the distribution and does not require a definition of the distribution parameters. By comparing the empirical
distributions, we can investigate how changes in the scenario characteristics modify the frequency of the maximum
inundation depths.
The study further looked at whether the depth of water at a station would change for various scenarios. Figure 6 shows
an example of the flood depth over simulated time at CI3 for the scenarios of Sandy. Pre-defined critical water levels
were investigated for each station, as hypothetical values representing the height between the floor and the critical
electric system in the station. Specifically, we considered 0.5 m, 1.5 m, and 2.5 m for threshold levels. As a measure
of the potential threat to the electric infrastructure, we determined the percentage of time that the flood level was over
each specific threshold (Figure 7). This data was then used to assess potential flooding problems associated with on-
site inundation: we associated the changes in risk posed to the CI from the different examined scenarios based on the
changes in those percentages.
**3 Results and Discussion**
**3.1 Flood extent**
The inundation extents shown in figure 6 represent an aggregation of the overall runs rather than a specific simulation
time, and it represents the extent reached when all pixels had the maximum inundation depth. Total flood extent ranged
between less than 1 km$^2$ to more than 7 km$^2$, with a minimum extent of 0.4 km$^2$ for the actual Sandy (SD1) at C8, and
a maximum extent of 7.1 km$^2$ for the future Sandy (SD5) at C3. The results showed consistent agreement that the
flood extent increased with increasing intensity of the event and an increase in the recurrence intervals of the flows
(Table 3).
Changes across the study sites relative to the FEMA 100-year flood extend (Table 4, Figure 7a–c) ranged from –87.8%
(for CI8 for SD1) to 192.2% (for CI2 for IR2). Overall, the sites with a return period of fewer than 100 years, showed
consistently less flooding than that of the FEMA map, a finding best represented by the comparison of actual events,
such as IR1.
Since the model performance shows a good agreement with the actual flood extents, and the HWMs (Chapt.2.2.3),
our results suggest that FEMA's flood maps do not fully capture the flood extent at least for some locations. Similar
findings were reported in Jordi et al. (2019), Wang et al. (2014) and Xian et al. (2005), where tens of meter-scale
absolute differences were found between the FEMA estimated flood extent for hurricane Sandy. The strength of
correlation (dCorr) between changes in the upstream (flow peak) or downstream (surge peak) components, and the
absolute differences with FEMA extent, gives an idea of the importance of each single driver of change. For the cases
investigated in this study, the percentage difference mostly depends on the surge: surge height explains more than
80% of the variation in the differences to FEMA extent (dcorr=0.8 in median). CI6 appears to be the sites where the
surge has the strongest correlation with the absolute difference in flood extent, as compared to FEMA maps. The
differences with FEMA maps are less related to the peak flows (median correlation 0.5, with max correlation recorded
for CI3). As expected, the correlation with surge increases at the decreasing of the hydrologic distance to the coast,
while the correlation with the flow increases the further a site is from the coast, even though this relationship is not
linear.
As we proceeded with the synthetic scenarios, adding compound and future climate, the results indicated the additional
impacts of the joint flood drivers (shifted tide, surge, SLR).
For the same event, peak storm-tide levels occurring near local high tide (i.e. SD2) resulted in more flooding than that
of events happening at low-tide (like actual Sandy, SD1). Climate change related SLR exacerbates extreme event
inundation relative to a fixed extent (FEMA) with variability that ranged from 8.3% (CI4/5) to as high as 425% (CI8).
CI8 is the site hydrologically closer to the coast (see hydrologic distance in Table 1), making it the most susceptible
to the altered scenario. Nonetheless, the shifted tide increased the inundation relative to the FEMA 100-year flood
map also for CI2 and CI4/5.
The effects of compound events emerged drastically with the combination of both shifted tide and SLR. With the
exception of CI3 and CI8, all other CIs showed an increase in the percentage change from FEMA (Table 4). In
comparison to SD1, SD3 exhibited increased inundation for all the CIs. The inundated area was about 146% more
$(1.9 \text{ km}^2)$ for SD3 than SD1 $(0.9 \text{ km}^2)$ for CI1, for example. The river flood peak for hurricane Sandy had a recurrence
interval of about two years, but the flood hazard associated with this event became more devastating if simulated in a
compound way, including SLR and shifted tide. This result suggests that events of lower river flood severity (from
less rain accumulations) can produce aggravating impact, as the intensity of major storm surges increases due to shifted
timing and SLR.
For the synthetic hurricane Florence and hurricane Irene, we saw an increased flooded area in comparison to FEMA
(Table 4); for CI2, for example, the increase was almost 200% from IR1 to IR2. Again, this result confirms that
accounting for river peak flow frequency alone does not effectively capture the severity of a flood hazard in the case
of coastal locations.
For all the study sites for future Sandy, we saw consistent increases in flood extent (Table 4) from SD2 to SD4 and
SD3 to SD5. Between SD2/SD3 and SD4/SD5, the only difference was the future projection of the flow. In comparison
to the FEMA map, the percentage change ranged from –22.3 to +123.7. CI1, CI7, and CI8 for SD4 have less inundation
than the FEMA 100-year map. This may be an indication of the significance of individual flood components specific
to one site. For those sites, river flow might not be the most significant component of the flood. When we look at the
hydrologic distances in table 1 CI1 and CI8 are closer to the coastline, making them more prone to coastal flooding
than fluvial flooding. When we looked at SD5 (which added SLR), all the sites except CI8 showed more flooding than
the FEMA 100-year flood map. Although CI8 had an increase of 22% in inundation compared to SD4.
When we compare the worst-case future events (SD5 and IR2) to actual events (SD1 and IR1), we can see major
changes in flood extents. The flood extent in all locations increased by about 60% on average for future Sandy with
both SLR and coinciding tide (SD5) in comparison to the actual Sandy (SD1), with the highest impact in CI8 (+148%).
Looking at Irene, the worst-case future scenario (IR2) increased the flood extent by about 30% on average for all
locations compared to the actual event (IR2), with the highest impact in CI2 (101%). Among all the events, Florence
had the lowest expected changes, between the current climate scenario (FL1) and the future one (FL2). One must note
that hurricane Florence had no actual impact in the study area; the simulation for this event was based on a hurricane
track forecast by GFS, which if materialized would have produced a flood inundation of almost 5 km$^2$ in CI3, and this
extent could have increased by about 20% in the worst-case future scenario (FL2) that includes shifted tide and SLR.
Five of the CIs were outside the FEMA 100-year flood zone, but they present flooding for FL1 and SD3. For FL2 all
of the study sites were more vulnerable (positive % change), as compared to the FEMA map. Similar findings are
presented for SD5, with the exception of CI8.

**3.2 Flood depths over the domain**
While flooding occurs in all the presented scenarios, both extent and depth vary greatly between the different
simulations. Depth is important to consider while preparing for risk management as it is used in determining flood
damage.
The CDFs of water depth for the whole domain (Figure 8), confirm that the water depths derived for coupled events
(i.e. high tide coinciding with surge peak, or SLR and future climate) are generally higher  than  those  derived from
events with independent drivers Note that for some cases (i.e. IR1 and IR2, for CI2 in Fig. 8) water depths increase
very consistently as  SLR increase.  Larges changes in the CDFs appears for lower water depths. Thus, regions with
generally lower hazard (depth), will likely experiences larger impacts under SLR. Results also confirm that scenarios
with simultaneous high values for all these parameters implicated a higher vulnerability of the CIs. Comparing these
changes in pairs [i.e. IR1 vs IR2, or SD1 vs SD3] also highlights that compound scenarios changes in the frequency
of extreme values that go far beyond the average are much more pronounced than the related changes of the median
depths (cumulative probability=0.50). In particular, it may be asserted that more expressed changes in extremes could
lead to corresponding "hazard shift" for all CIs, as represented in Figure 8.

These results suggest that fluvial flow is not the only driver determining flood risk. Actual Irene (IR1) and synthetic
Florence (FL) had higher river flood return periods than did actual Sandy (SD1) (Table 2). Nonetheless, the CDFs of
the flood depth showed different behavior in terms of severity. For CI1, for example, IR1 had higher probabilities for
lower depth, followed by SD1 and FL1. In CI8, SD1 had higher probabilities for lower values of depth. These findings
highlight that neither the severity of rainfall, nor the magnitude of river flow controls the flood characteristics, which
are, rather, controlled by additional factors, such as storm surge, high tides, topography, and location of the site. CI7,
for example, which is more coastal than the other CIs, presented increasing flood depth due to tidal timing.
As expected, and as previously highlighted when considering the flood extent (Table 4), climate played an important
role in flood hazard changes. Furthermore, the effect of SLR was also evident for all the events (IR, SD, and FL),
increasing the flood depth for the same exceedance probability. For CI6, for example, the 50% exceedance
corresponded to ~1 m depth of floodwater for IR1, increasing to ~1.5 m for IR2. For the CI4 and CI5 sites, for
exceedance of 20%, actual Irene produced ~2 m of flood depth, whereas with SLR it was ~2.5 m. Another way to put
it is that, for CI4/5, IR1 had an exceedance of ~20% for a flood depth of 2 m, whereas IR2 had an increased exceedance
level of 40%. Similarly, for 50% exceedance, FL1 and FL2 corresponded to 1.5 m and 2 m depth of floodwater,
respectively, and we saw the trend for the Sandy event scenarios (SD2–SD3; SD4–SD5) as well.
This analysis highlighted that the timing of a storm is also crucial. The changes from SD1 to SD2 showed very well
the impact of the shifted tide for all the sites. For CI3, for example, the 1 m flood depth had an exceedance of ~88%
for SD2, whereas it was only ~23% for SD1.
Analysis of the overall flood depth across the whole domain shows that the coincidence of fluvial flood, high tide, and
storm surge results in a significant increase in flood risk. SD3 and SD5 had all the components of a compound flood
and comparing them with SD1 gave us a clear idea of how severe a compound event can be in the future. CI3, for
example, had exceedance levels of almost 30%, 85%, and 90%, respectively, for SD1, SD3, and SD5 for a flood depth
of 1 m. This suggests the compound effect increases the intensity of the flood hazard.
**3.3 Local risk for CI**
Much of the flood damage in CI is incurred by components being submerged for a long period. Investigating the
duration of the flood depth at the CI location (Figure 9) should be considered in planning for any protective measures,
such as elevating or waterproofing equipment. If a critical infrastructure shows 0%, it means that for that
scenario/event the water didn't reach the substation at all, at least during the simulated timeframe. This could be due
to the water flooding other upstream locations, and therefore draining away from the station, or because the topography
of the landscape actually prevented water from reaching the area for some specific events.
According to our analysis, none of the scenarios has an actual impact on CI1. For the other CIs, comparing individual
events we could see an increase in risk due to the compound hazard scenarios—that is, shifted tide and SLR. Important
to note is that, for most of the sites, the compound risk due to SLR and tide timing was always higher for the lower
water-level thresholds (0.5 m). This implies a higher risk for CI components currently positioned closer to the ground.
Damage to the CI components is dictated by both the flood depth and the duration of submergence. The suggested
high values of risk [increase percentage in inundation duration] (Figure 9) further imply differences in the timing of
repairs.  In the cases of CI7 and CI8 (Figure 9), the CIs remained submerged with 0.5 m of water for about 20% of the
event period for actual Sandy, but for the worst-case future Sandy scenario, the location was flooded for more than
90% of the event duration. This demonstrates the increased flood risk to which future climate conditions expose CI.
Another important insight was provided by the hurricane Florence scenarios. As mentioned earlier, Florence did not
affect the study area, although an early GFS storm forecast track predicted landfall in Long Island and Connecticut.
For this event, the estimated measure of risk was about 20%, and it was shown to increase to up to 40% for the lower
water depth (0.5 m) threshold in some locations. The result of the simulated scenario allows for an assessment of
potential damage and for an identification of equipment that might be affected by future events under current climatic
conditions. In this regard, comparing the results for the different CIs during the Sandy scenarios revealed an interesting
pattern. While we might have expected a greater impact over the whole domain when shifting the tide (Figure 9, Table.
3), we found different impacts in the CI locations.  Notably, the risk appeared lower when the tides were shifted (Fig.
9) for some of the CIs (for example, CI5 and CI7). This can be explained by the fact that higher water levels in the
domain were changing the water flows, allowing the flood to follow different drainable ways. This can be a very
useful piece of information for deciding whether to and where to take measures in terms of flood occurrence and
potentially relocating CIs to avoid catastrophic compound flood events.
From table 1 we can see that CI8 is the closest to the coastline followed by CI7, CI6, and CI5. From figure 9 we can
see that all the CIs that are closer to the coastline are susceptible to changes in the downstream water level condition
(Shifted tide/ SLR) (Table 3). CI4 is the farthest from the coast followed by CI3. Both the CIs show minimal response
to changes in the coastal water level compared to CI5/ CI6/ CI7. This analysis gives us conclusive evidence of risk
associated with the location of the CI from the coastline.

## 4 Concluding Remarks

Preparing for the challenges posed by climate change requires understanding of current actual, possible and future
scenario of tropical storm impacts, and a correct understanding of the hazard imposed by compound flooding.  In this
work we have developed and implemented a modeling framework that allows to address this task, focusing on coastal
electric grid infrastructure (substations). To date, the design of these facilities typically has assumed the current
climatic conditions. However, a changing climate, as well as co-occurrence of compound drivers, and the resulting
more extreme weather events mean those climate bands are becoming outdated, leaving infrastructure operating
outside of its tolerance levels.
We explored a range of actual and synthetic hurricane scenarios, offering a system that could inform short- and long-
term decisions. For the short-term decision, the framework allowed to investigate the characteristics of the hurricane-
related inundation, considering the compound effect of riverine and coastal flooding coinciding, or not, with peak high
tides. Generally, hurricanes affect large areas, and the specific locations at which damage will occur are often difficult
to anticipate. Simulation of different scenarios can provide system operators with the ability to prepare for damage
and respond quickly once it has occurred—for example, by pre-positioning repair crews. Furthermore, by simulating
the impact using possible storm paths, the framework allows us to understand the potential impacts on the CI. The
framework proposed in this study evaluates the extent of flood nearby a critical coastal infrastructure caused by
possible extreme compound events. Each type of infrastructure system has specific elements vulnerable to specific
water levels; we map those hazard infrastructure intersections where risks will be exacerbated by climate change or
compound events.
The findings of this study can support flood mitigation; the FEMA 100-year map is used for designing infrastructure
and for making decisions on flood mitigation and flood insurance. Nonetheless, these maps must be updated because
flood risk is not static; changes in hydrology, topography, and land development all have an impact on flood
conditions. The results show that the vulnerability of each substation is linked to the different storms' characteristics,
and how they vary depending on the distance from the coast—that is, inland substations are less affected by surge and
SLR and more affected by rainfall accumulation events (such as Irene).  The findings of this study highlight that rising
seas will allow storm surges to inundate areas farther inland and that flood hazard is likely to grow as seas rise and
storm surges become deeper. The results also highlight that tide-surge-SLR effects modeled using only coastal models
in isolated open environments without considering fluvial effects on the flooding, or riverine models without
appropriate downstream boundary conditions cannot capture the risk from tide-surge-SLR effects. The variability in
flood extent among scenarios implies that the modeling of individual flood drivers separately can mischaracterize the
true risk of flooding to coastal communities and critical infrastructure, introducing uncertainties that make the design
of long-lived infrastructure much more difficult. Significant losses can result in when the designs are inadequate and
ill-adapted to climate conditions.
This study also shows that, for some locations, FEMA maps significantly underestimate the actual storm surge risk to
structures near the shore relative to structures further inland, and it generally does not account for the impacts posed
by simultaneous conditions, such as high tide and river flows, or for future climate impacts.
The inundation maps, as well as the depth distributions, highlight how climate change is expected to lead to increased
flooding in many sites, due to rising sea levels and changing precipitation patterns. The impacts will be felt most
acutely along the coasts, but our results show a significant increase also for the more inland locations, as heavy and
more frequent rain events increase the risk of flash floods and riverine flooding events. The provided framework can
produce inundation maps that would allow improving the CIs' resiliency in the face of natural disasters, independently
from the mapping done for insurance purposes. Critical infrastructures are usually positioned by following the FEMA
100-year flood zone map. Areas outside the designated zones generally either do not have flood mitigation plans, and
stand without any protection, or plans based on critical flood depths derived from FEMA zone areas. In this study,
however, we see an increase in the exposed (flooded) areas for future climate scenarios, as well as some under- and
overestimation as compared to FEMA maps. We also show how the flood depth exceedance probability at a location
can essentially increase during compound flooding and shift due to climate changes. This further suggests the need to
develop, update improved criteria for recognizing the effects of existing and planned protection measurements, such
as relocating equipment or Cis, where warranted.
Future research should consider improved estimation methods, including more detailed information on the variability
of river properties (i.e. depth and width). Future works should also relate the frequency of inundation depths to return
periods of precipitation, river flows, and surges, as well as differentiate among the individual effects of the components
to determine the role of each in flooding impact. This can be a very useful piece of information for deciding whether
and where to take measures in terms of flood occurrence and the potential relocation of CI to avoid catastrophic
compound flood events.
Notwithstanding these challenges, the findings of this study highlight that, whenever possible, risk assessments across
different critical locations directly or indirectly affecting critical infrastructure should be based on a consistent set of
compound risks. The proposed analysis suggests planning and management strategies for critical infrastructure should
rely on historical flooding data, together with future storm scenarios and climate and SLR projections. The overall
impact on each critical structure in terms of flood extent and depth is unique. This will ultimately allow the building
of resilience into different components of critical infrastructure to enable the system to function even under disaster
conditions or to recover more quickly.

**Acknowledgments**: This work was supported by Eversource Energy.
**Author contributions:** MKh, GS, XS, EA conceived the study. XS and EA contributed to the conception of the
hydrologic model. RL contributed to the production and analysis of the hydrologic model outputs. MKo and EN
contributed to the analysis, and interpretation of the climatic data. MKh and GS contributed to the automation of the
hydraulic model and the interpretation of its results. All authors participated in drafting the article and revising it
critically for important intellectual content. All authors give the final approval of the published version.
**Competing interests.** The authors declare that they have no conflict of interest.

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

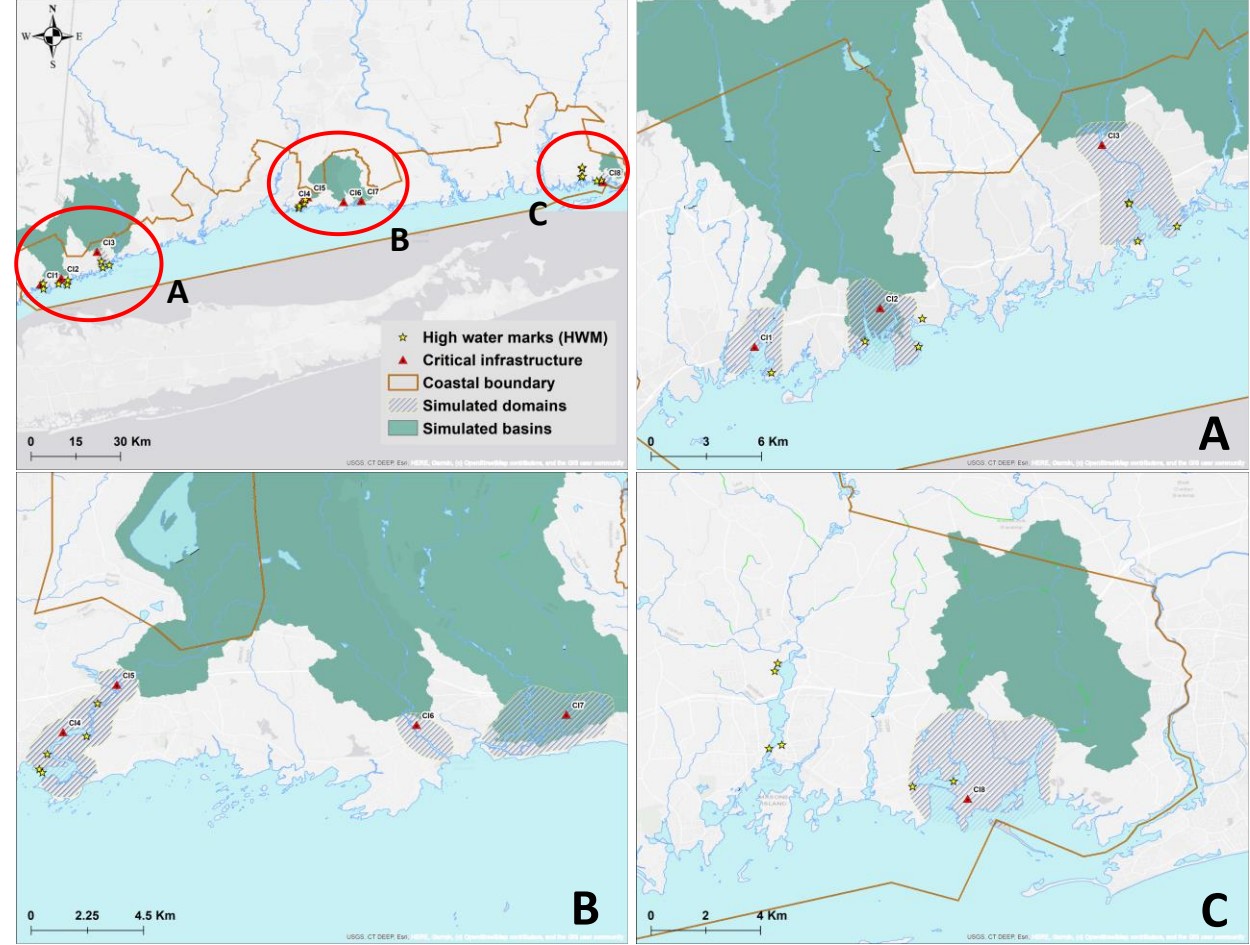


**Figure 1: Study area with associated watersheds and simulation domains. Locations of substations and USGS high water marks are also shown. Red circles in the top left-hand panel, and marked with A, B, and C are highlighted in the panels A to C respectively. Background map by ESRI web-services, provided by UConn/CTDEEP, Esri, Garmin, USGS, NGA, EPA, USDA, NPS**

| Atmospheric Simulation | Hydrologic Modeling | Hydrodynamic Modeling |
|---|---|---|
| Inputs:<br>• GFS Forecasts (Florence)<br>• ECMWF ERA-Interim-GCM AR4 A2 (Future Sandy)<br>• USGS GMTED2010 30-arc-sec<br>• Noah-modified IGBP-MODIS.<br>• Hybrid STATSGO/FAO (30sec).<br>Outputs:<br>• Metorologic forcing: precipitation, short and longwave radiation, specific humidity, air temperature, air pressure and wind speed | Inputs:<br>• Forcing from NLDAS(1979-) (Sandy, Irene)<br>• Forcing from WRF (Florence, Future Sandy)<br>• DEM (USGS, 2016)<br>• LULC (MOD12Q1 from MODIS)<br><br>Output:<br>• Streamflow of historic events | Inputs:<br>• Lidar 2016 (1m)<br>• NLCD 2011<br>• Building footprints, 2012<br>• Streamflow from hydrologic model<br>• NOAA tide & surge<br>• Sea level rise (0.5 m for 2050)<br><br>Outputs:<br>• Flood risk<br>• Inundation and flood depth<br>• Scenario based hazard assessment |


**Figure 2: Considered framework including atmospheric simulations, hydrologic, and hydrodynamic modeling. Hurricane**
**events (actual and simulated), and inputs and outputs of each component are shown. Readers should refer to chapter 2.2**
**for specifications**

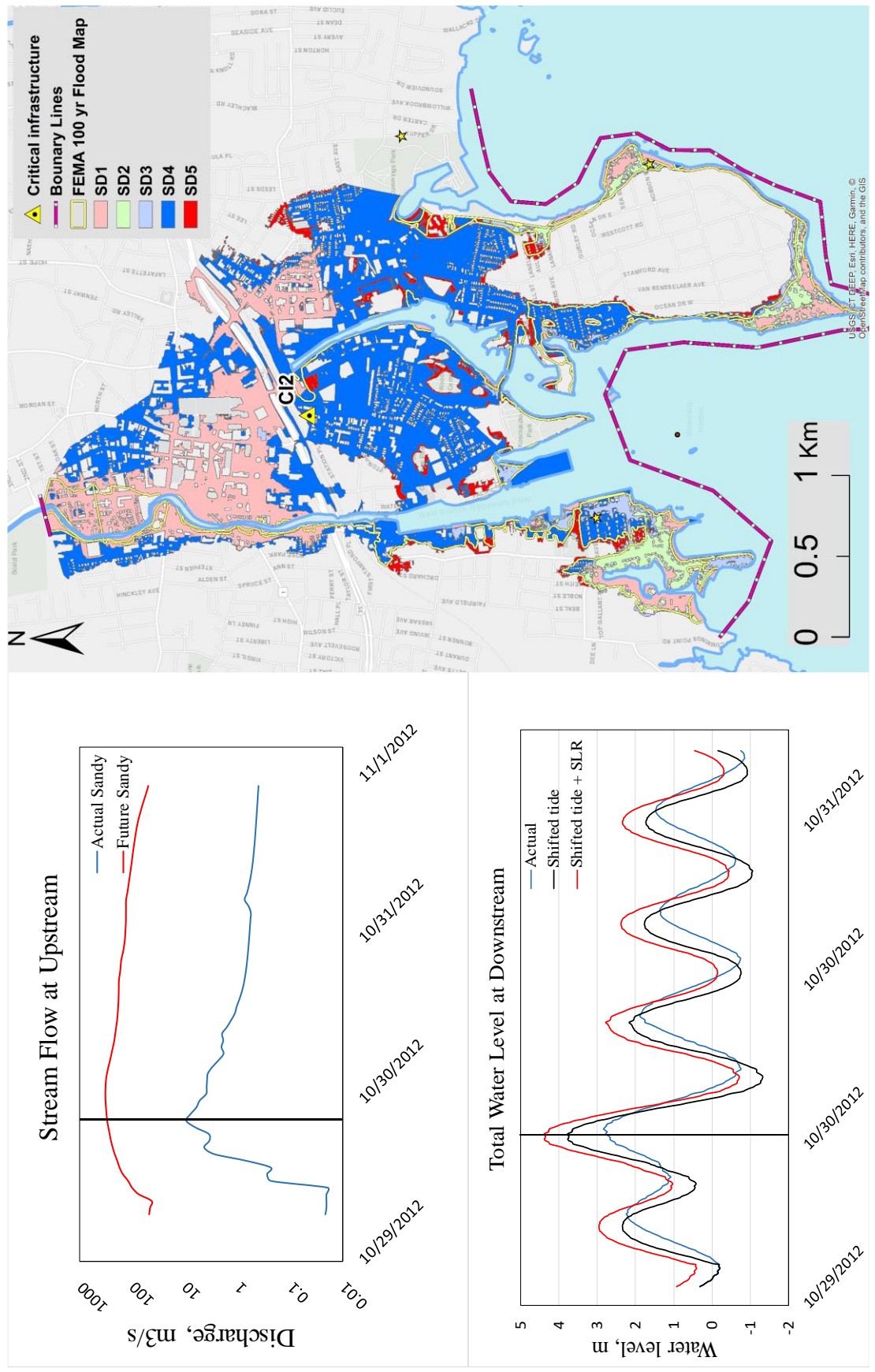

**Figure 3: Example of different scenarios showing the upstream boundary condition (top left-hand panel, including the discharge for actual Sandy and future Sandy), and downstream boundary (bottom left-hand panel, including tide, shifted tide, and shifted tide with SLR). Output flood extend is also shown (right-hand panel), including results for SD1 to SD5 [reader should refer to Tab. 3 and chapter 2.2 for specification on the scenarios]). Background map on the firhg-hand panel by ESRI web-services, provided by UConn/CTDEEP, Esri, Garmin, USGS, NGA, EPA, USDA, NPS**

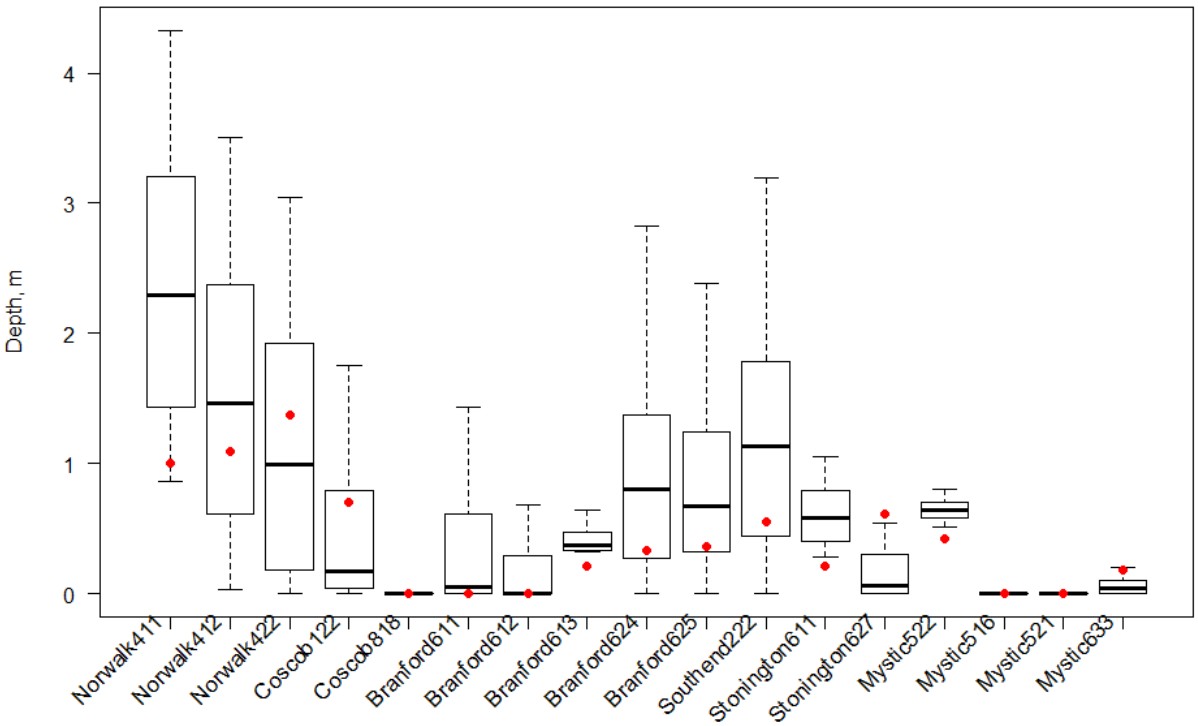

**Figure 4: Validation results (boxplot of water depth within 10x10m around the high-water mark -HWM- location) compared to selected HWM (red dots) by USGS**

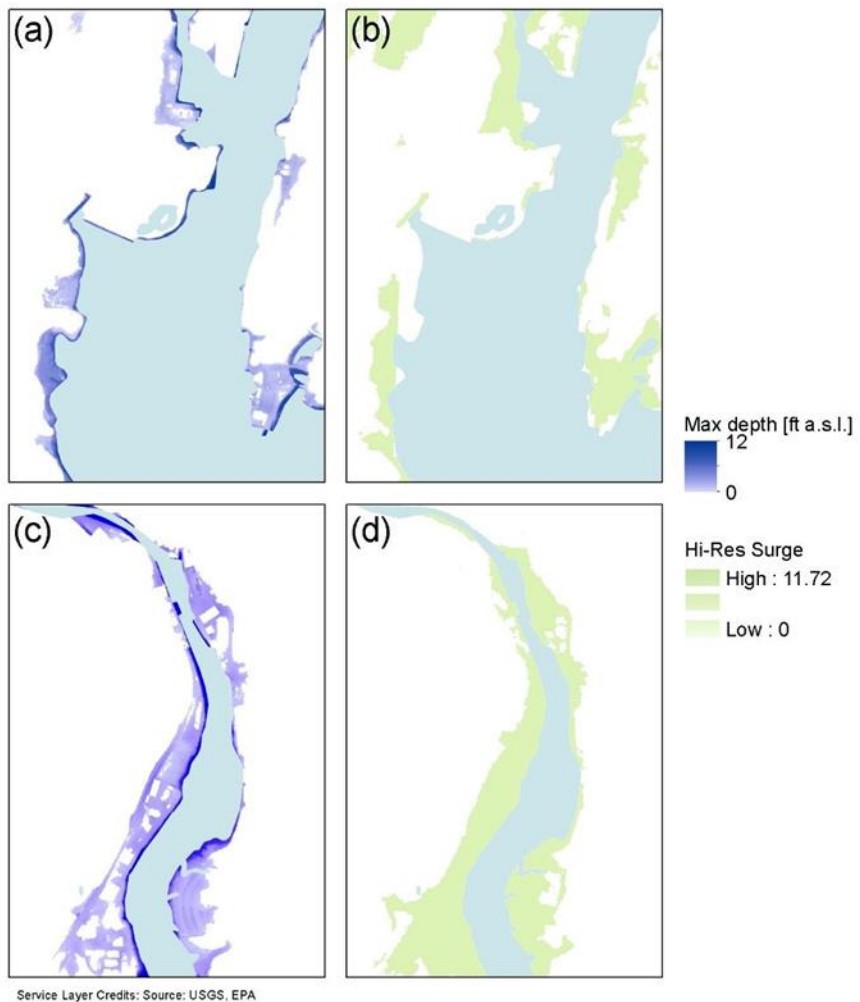

Service Layer Credits: Source: USGS, EPA
FEMA, CT DEEP

**Figure 5: Comparison between the results of the proposed model for two selected locations (a,c, CI1 and CI2 respectively) and the maximum surge extent as proposed by CtEco (c,d respectively).**

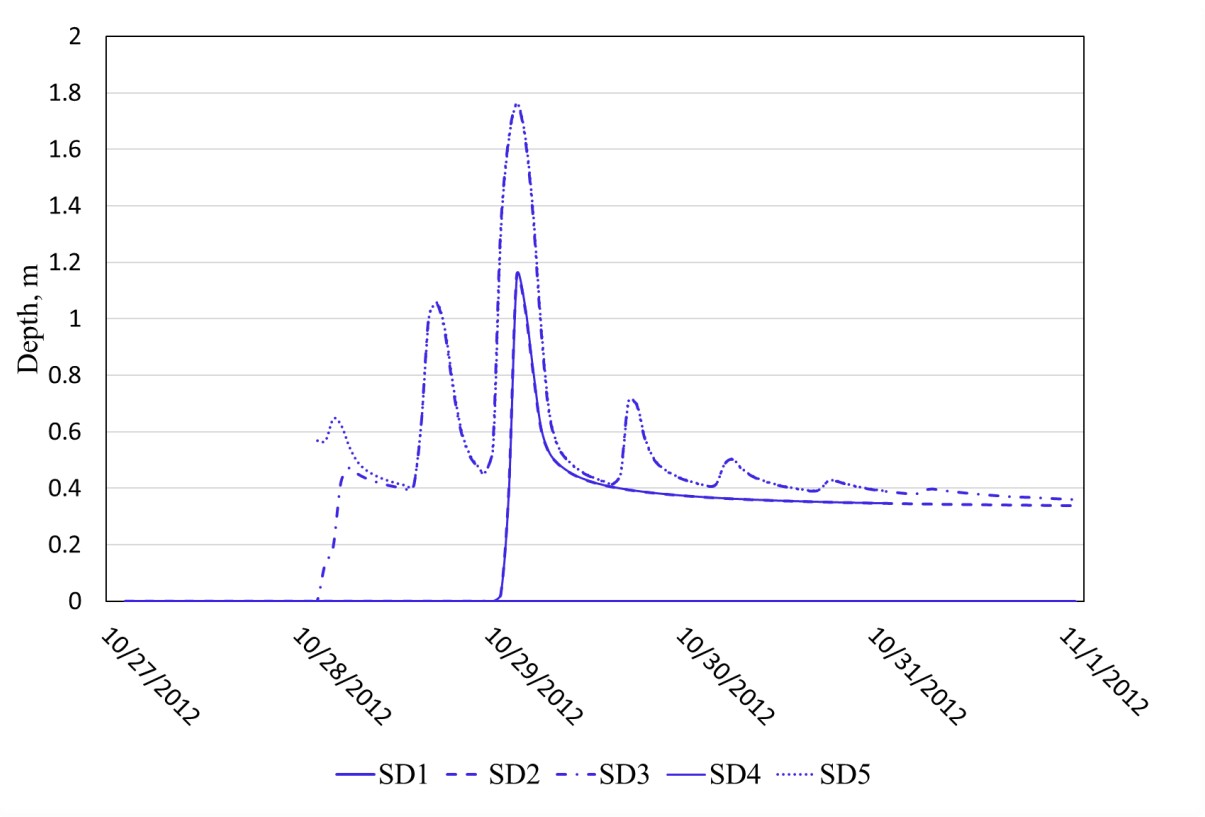

**Figure 6: Example of time series of depth values for the different scenarios of Sandy event at CI3 [SD1 to SD5, readers should refer to Table 3 and chapter 2.4 for specification on the scenarios]**

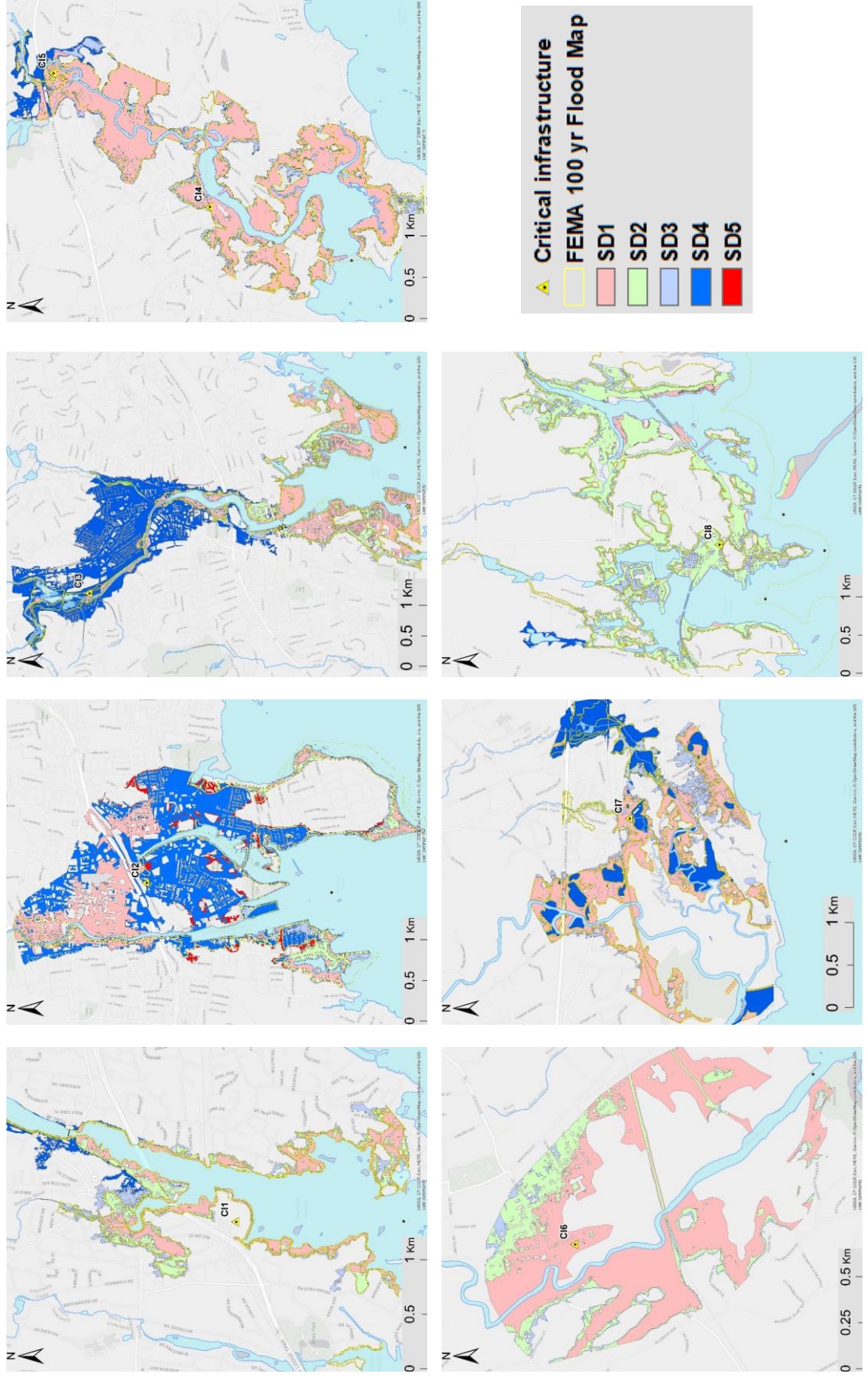

**Figure 7a: Map overlay of maximum inundation for all the study domains containing CI1 through CI8 for the scenarios of Sandy [SD1 to SD5, readers should refer to Table 3 and chapter 2.2 for specification on the scenarios]**

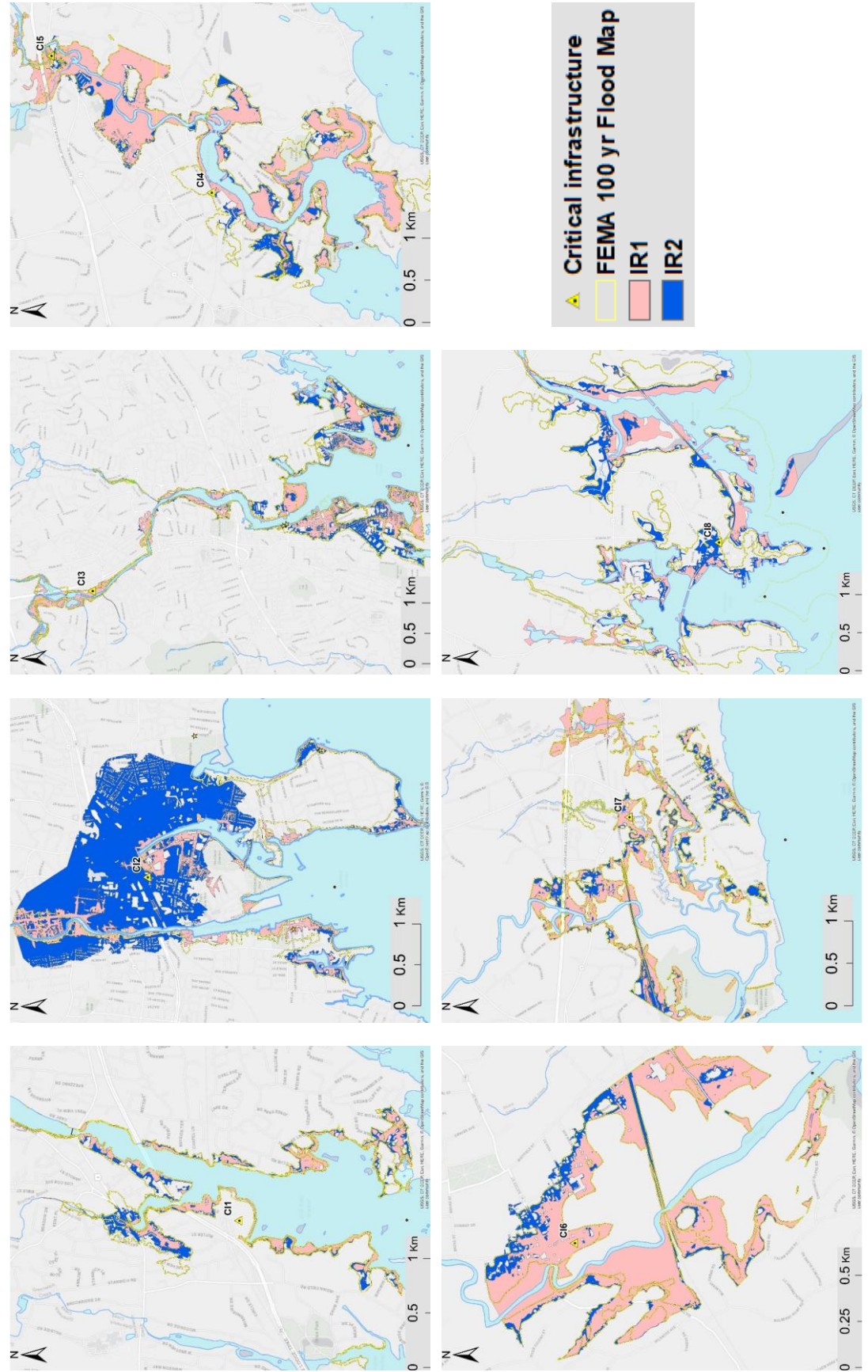

**Figure 7b: Map overlay of maximum inundation for all the study domains containing CI1 through CI8 for the scenarios of Irene [IR1 and IR2, readers should refer to Tab. 3 and chapter 2.2 for specification on the scenarios]. Background map by ESRI web-services, provided by UConn/CTDEEP, Esri, Garmin, USGS, NGA, EPA, USDA, NPS**

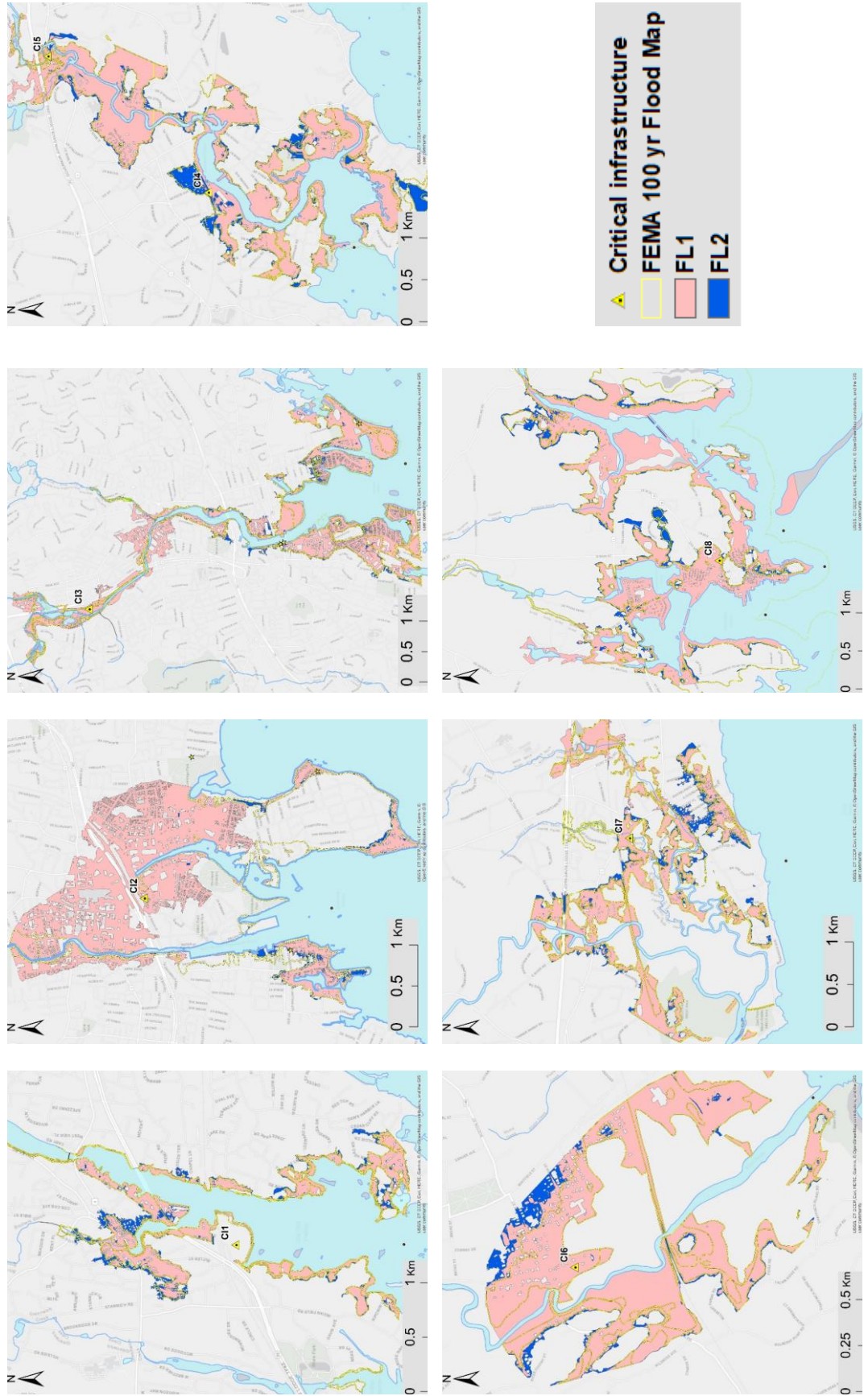

**Figure 7c: Map overlay of maximum inundation for all the study domains containing CI1 through CI8 for the scenarios of Florence [FL1 and FL2, readers should refer to Table 3 and chapter 2.2 for specification on the scenarios]. Background map by ESRI web-services, provided by UConn/CTDEEP, Esri, Garmin, USGS, NGA, EPA, USDA, NPS**

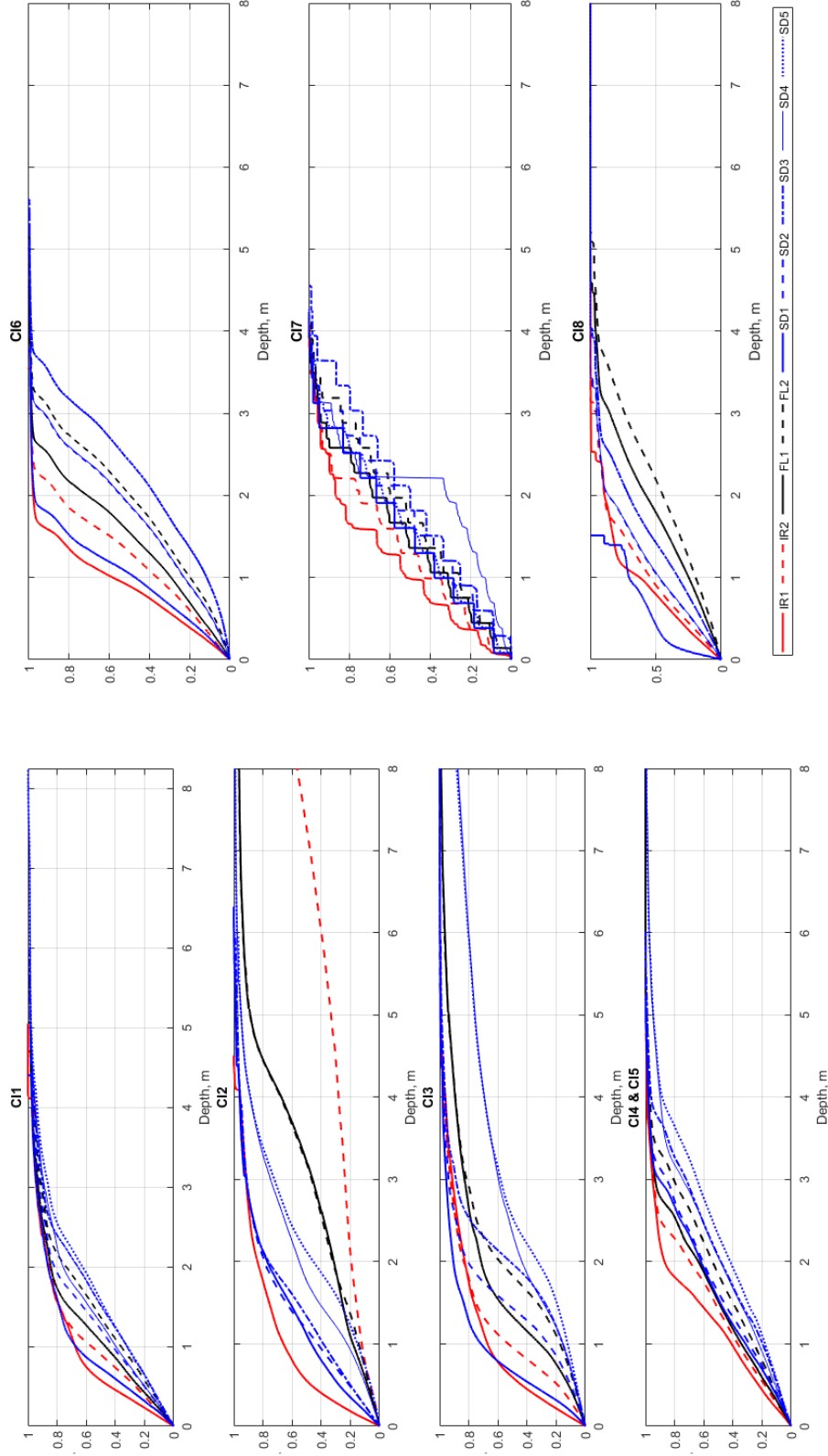

**Figure 8: Cumulative density plot of the depth of all the flooded cells during maximum inundation. Hurricanes scenarios are labelled according to Table 3 and explained in chapter 2.2. Critical infrastructures are labelled CI1 to CI8, as described in Table 1.**

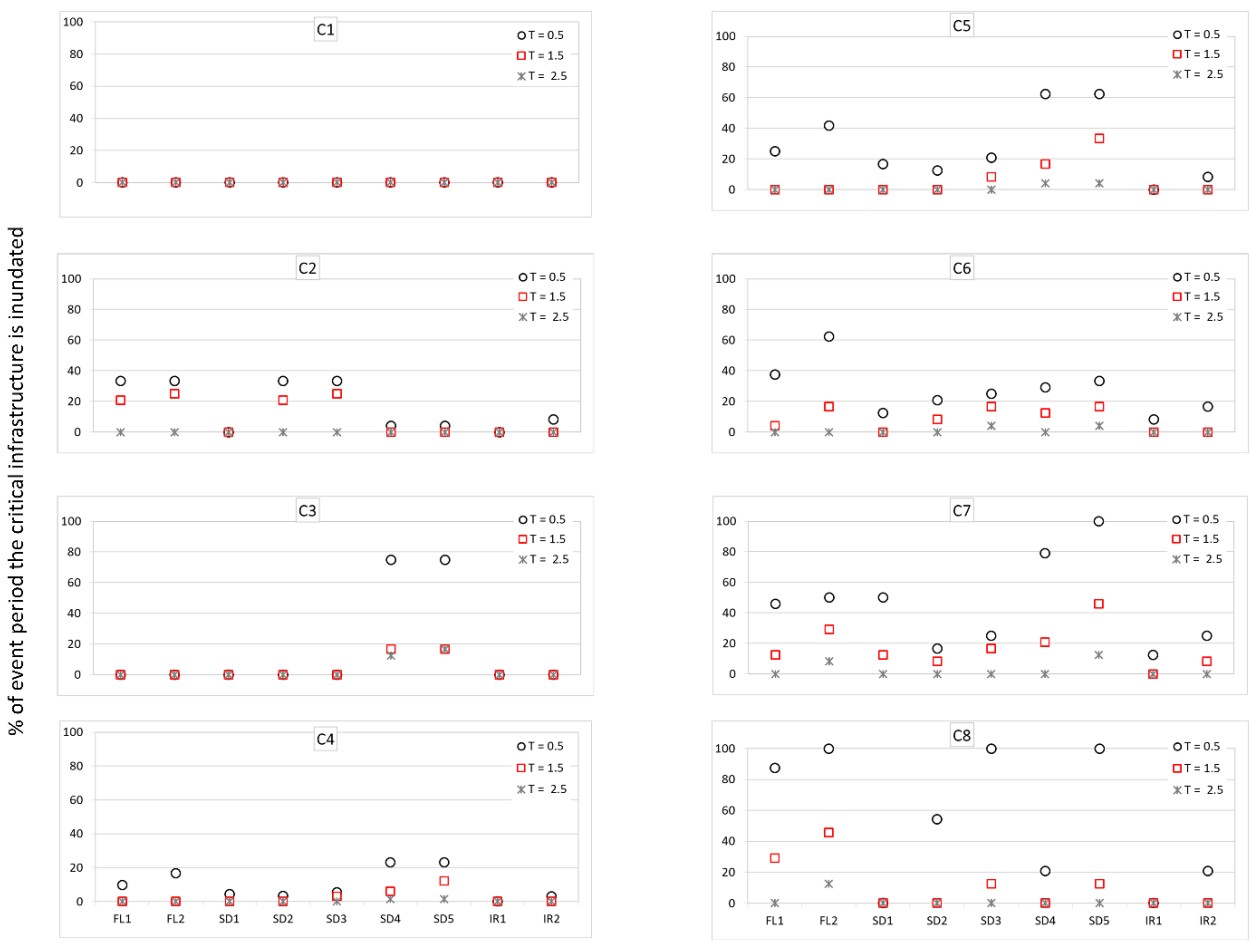

**Figure 9: Peak over threshold (T=0.5, 1.5 and 2.5m) at selected critical infrastructures. Hurricanes scenarios, along the x-axis, are labeled according to Table 3 and explained in chapter 2.2. Critical infrastructures are labeled CI1 to CI8, as described in Table 1.**

**Table 1: Study area- Characteristics of the considered CIs, with river and model domain information. Basin area represents the area of the underlining watershed; domain area is the extent of the simulation domain; reach length represents the length of the stream within the domain; hydrologic distance represents the distance from each CI to the coastline.**

| Critical Infrastructure (CI) | Town | Rivers | Basin area, $km^2$ | Domain area, $km^2$ | Reach length, km | Hydrologic distance, km |
|---|---|---|---|---|---|---|
| CI1 | Coscob | Mianus River | 216.6 | 7.5 | 7.8 | 4.5 |
| CI2 | Southend | Rippowam River | 308.4 | 12.1 | 4.9 | 5.3 |
| CI3 | Norwalk | Norwalk River | 268.7 | 20.7 | 8.3 | 7.8 |
| CI4/ CI5 | Branford | Branford River | 84.5 | 7.9 | 6.7 | 8.8/5.3 |
| CI6 | Guilford | West River | 126.4 | 2.2 | 3.7 | 5.1 |
| CI7 | Madison | East & Neck Rivers | 173.0 | 8 | 5.3 | 6.8 |
| CI8 | Stonington | Stonington harbor | 10.0 | 14.9 | 5.2 | 2.9 |

**Table 2: Model domain information for Florence**

| | |
|---|---|
| Horizontal Resolution | 18, 6, and 2 km |
| Vertical levels | 28 |
| Horizontal Grid Scheme | Arakawa C grid |
| Nesting | Two-way nesting |
| Convective parameterization | Grell 3D ensemble scheme (18 and 6 km grids only) |
| Microphysics option | Thompson graupel scheme (Thompson et al., 2008) |
| Longwave Radiation option | RRTM scheme (Mlawer et al., 1997) |
| Shortwave Radiation option | Goddard Shortwave scheme (Chou and Suarez 1994) |
| Surface-Layer option | Monin-Obukhov Similarity scheme |
| Land-Surface option | Noah Land-Surface Model (Tewari et al., 2004) |
| Planetary Boundary Layer | Yonsei scheme (Song–You et al., 2006) |

**Table 3: Peak Tide, Surge at the maximum total water level instance, Accumulated precipitation & peak flows (with return period reported within brackets) for the simulated scenarios. Reader should refer to Chapter 2.2 for a detailed description of each hurricane scenario (IR for Irene, SD for Sandy, FL for Florence). The "*" denotes the scenarios having sea level rise (SLR) added to the surge. Critical infrastructures are labelled CI1 to CI8 according to Table 1.**

| Scenarios | | CI1 | CI2 | CI3 | CI4/CI5 | CI6 | CI7 | CI8 |
|---|---|---|---|---|---|---|---|---|
| FL1 | Tide (m) | 0.99 | 0.99 | 0.99 | 0.94 | 0.94 | 0.94 | 0.17 |
| | Surge (m) | 2.51 | 2.51 | 2.51 | 2.56 | 2.46 | 2.56 | 3.33 |
| | Accumulated precipitation (mm) | 128.5 | 147.5 | 165.1 | 192 | 203.9 | 200.7 | 289.2 |
| | Peak flow, m3/s (return period) | 51.3 (<2) | 87.4 (5) | 74.9 (<2) | 106.1 (13) | 113.3 (8) | 143.2 (51) | 93.1 (6) |
| FL2* | Tide (m) | 0.99 | 0.99 | 0.99 | 0.94 | 0.94 | 0.94 | 0.17 |
| | Surge (m) | 3.12 | 3.12 | 3.12 | 3.17 | 3.07 | 3.17 | 3.93 |
| | Accumulated precipitation (mm) | 128.5 | 147.5 | 165.1 | 192 | 203.9 | 200.7 | 289.2 |
| | Peak flow, m3/s (return period) | 51.3 (<2) | 87.4 (5) | 74.9 (<2) | 106.1 (13) | 113.3(8) | 143.2 (51) | 93.1 (6) |
| SD1 | Tide (m) | 0.82 | 0.82 | 0.82 | 0.4 | 0.4 | 0.4 | 0.01 |
| | Surge (m) | 2.37 | 2.37 | 2.37 | 2.3 | 2.3 | 2.3 | 1.87 |
| | Accumulated precipitation (mm) | 24.8 | 24.7 | 21.5 | 17 | 17.7 | 15.1 | 8.9 |
| | Peak flow, m3/s (return period) | 3.4 (<2) | 9.3 (<2) | 3.3 (<2) | 4.7 (<2) | 1.3 (<2) | 0.9 (<2) | 0.03 (<2) |
| SD2 | Tide (m) | 1.01 | 1.01 | 1.01 | 1.13 | 1.13 | 1.13 | -0.15 |
| | Surge (m) | 2.56 | 2.56 | 2.56 | 2.8 | 2.8 | 2.8 | 1.95 |
| | Accumulated precipitation (mm) | 24.8 | 24.7 | 21.5 | 17 | 17.7 | 15.1 | 8.9 |
| | Peak flow, m3/s (return period) | 3.4 (<2) | 9.3 (<2) | 3.3 (<2) | 4.7 (<2) | 1.3 (<2) | 0.9 (<2) | 0.03 (<2) |
| SD3* | Tide (m) | 1.01 | 1.01 | 1.01 | 1.13 | 1.13 | 1.13 | -0.15 |
| | Surge (m) | 3.12 | 3.12 | 3.12 | 3.4 | 3.4 | 3.4 | 2.564016 |
| | Accumulated precipitation (mm) | 24.8 | 24.7 | 21.5 | 17 | 17.7 | 15.1 | 8.9 |
| | Peak flow, m3/s (return period) | 3.4 (<2) | 9.3 (<2) | 3.3 (<2) | 4.7 (<2) | 1.3 (<2) | 0.9 (<2) | 0.03 (<2) |
| SD4 | Tide (m) | 1.01 | 1.01 | 1.01 | 1.13 | 1.13 | 1.13 | -0.15 |
| | Surge (m) | 2.56 | 2.56 | 2.56 | 2.8 | 2.8 | 2.8 | 1.95 |
| | Accumulated precipitation (mm) | 555.3 | 546.9 | 526.8 | 338.2 | 330.2 | 316.6 | 323.7 |
| | Peak flow, m3/s (return period) | 242.4 (316) | 319.1 (326) | 201.7 (28) | 178.3 (98) | 168.4 (48) | 197.0 (301) | 94.7 (6) |
| SD5* | Tide (m) | 1.01 | 1.01 | 1.01 | 1.13 | 1.13 | 1.13 | -0.15 |
| | Surge (m) | 3.12 | 3.12 | 3.12 | 3.4 | 3.4 | 3.4 | 2.564016 |
| | Accumulated precipitation (mm) | 555.3 | 546.9 | 526.8 | 338.2 | 330.2 | 316.6 | 323.7 |
| | Peak flow, m3/s (return period) | 242.4 (316) | 319.1 (326) | 201.7 (28) | 178.3 (98) | 168.4 (48) | 197.0 (301) | 94.7 (6) |
| IR1 | Tide (m) | 1.16 | 1.16 | 1.16 | 1.1 | 1.1 | 1.1 | 0.93 |

|  | Surge (m) | 1.94 | 1.94 | 1.35 | 1.42 | 1.42 | 1.42 | 1.1 |
|---|---|---|---|---|---|---|---|---|
|  | Accumulated precipitation (mm) | 187.8 | 177.8 | 173.5 | 98.1 | 91.6 | 86.1 | 58.5 |
|  | Peak flow, m3/s (return period) | 158.5 (56) | 201.1 (58) | 126.7 (26) | 93.9 (5) | 85.7 (5) | 93.5 (5) | 30.8 (3) |
|  | Tide (m) | 1.16 | 1.16 | 1.16 | 1.1 | 1.1 | 1.1 | 2 |
| IR2* | Surge (m) | 2.54 | 2.54 | 1.94 | 2.03 | 2.03 | 2.03 | 1.7 |
|  | Accumulated precipitation (mm) | 187.8 | 177.8 | 173.5 | 98.1 | 91.6 | 86.1 | 58.5 |
|  | Peak flow, m3/s (return period) | 158.5 (56) | 201.1 (58) | 126.7 (26) | 93. 9(5) | 85.7 (5) | 93.5 (5) | 30.8 (3) |

**Table 4: Overall extent of the inundated area (in km$^2$), the relative difference (% change in parenthesis) compared to the FEMA 100yr Flood Zone and dCorr (correlation between differences in flood extent as compared by FEMA, and flow and surge peak)**

| CIs | FL1 | FL2 | SD1 | SD2 | SD3 | SD4 | SD5 | IR1 | IR2 | dCorr surge | dCorr flow |
|---|---|---|---|---|---|---|---|---|---|---|---|
| CI1 | 1.6 (-8.5) | 1.8 (2.9) | 0.9 (-48.1) | 1.4 (-21.7) | 1.9 (8.3) | 1.7 (-2.8) | 2.0 (13.9) | 1.3 (-27.5) | 1.5 (-15.9) | 0.86 | 0.40 |
| CI2 | 3.9 (134.2) | 4.0 (139.4) | 1.9 (-12.7) | 2.1 (25.6) | 2.3 (36.3) | 3.7 (123.7) | 4.8 (185.2) | 1.6 (-1.9) | 4.9 (192.2) | 0.53 | 0.55 |
| CI3 | 4.7 (2.6) | 4.9 (7.5) | 3.5 (-24.5) | 4.0 (-10.5) | 4.3 (-6.2) | 5.4 (17.5) | 7.1 (56.2) | 3.2 (-29.3) | 4.0 (-12.1) | 0.67 | 0.70 |
| CI4/CI5 | 2.7 (-8.3) | 3.2 (8.4) | 2.4 (-18.5) | 2.6 (0.3) | 3.4 (13.8) | 2.9 (2.5) | 3.6 (22.2) | 2.0 (-32.3) | 2.4 (-17.3) | 0.98 | 0.43 |
| CI6 | 0.9 (3.7) | 0.9 (13.1) | 0.7 (-14.9) | 0.8 (-10.3) | 1.0 (16.6) | 0.9 (11.4) | 1.0 (16.5) | 0.7 (-20.4) | 0.8 (-4.8) | 0.84 | 0.56 |
| CI7 | 2.5 (1.0) | 2.7 (12.5) | 1.6 (-33.9) | 2.0 (-12.8) | 2.6 (8.5) | 2.1 (-10.7) | 2.6 (7.3) | 1.9 (-23.5) | 2.3 (-7.5) | 0.81 | 0.46 |
| CI8 | 3.1 (4.5) | 3.5 (18.4) | 0.4 (-87.8) | 2.1 (-28.8) | 2.6 (-11.1) | 2.2 (-22.3) | 2.7 (-8.9) | 1.1 (-63.1) | 1.8 (-37.9) | 0.88 | 0.67 |

Note: (-) Area inundated less than FEMA's 100yr zone