# Peer review of "Impact of Compound Flood Event on Coastal Critical Infrastructures Considering Current and Future Climate"

_Natural Hazards and Earth System Sciences, 2020_

## Referee Comment (RC1) · Anonymous Referee #1 · 3 Aug 2020

This paper studies how compound flood hazards can affect critical infrastructure in coastal Connecticut. It is a good attempt to quantitatively evaluate the changes of flood risk for different compound scenarios, which provides a good perspective to investigate the potential impacts of flooding on critical infrastructures such as substations when considering multi flood drivers. However, the analysis of the results are not thorough and some descriptions are quite confusing, especially in the part of discussions. Besides, there are some obvious mistakes and typos in this paper. For publishing in NHESS, this paper in its current form requires major revisions and editing. Detailed comments are listed below: Comments: 1. The literature review for frequency-based effect estimate of compound-event flooding (Line 69-76) is obscure. What is the missing link in the current research and why most of the studies failed to or avoided to

explore the frequency and risk assessment of the compound flooding? It seems in this study the authors designed several compound scenarios to consider the probability of precipitation and surge as a solution to the shortcoming associated with compound flood risk assessment. If this is the case, more details on the related theories and methodologies should be presented in the introduction. 2. In section 2.3, it is better to use a table to describe these compound scenarios and their related hurricanes, SLR, tide conditions and other attributes. 3. In section 2.4, which site does Figure 5 present for? The red rectangle shows a window of 48 hrs, not 24 hrs. What criterion is used for selecting the window size? 4. In Lines such as 230, 237, Table 4 should be Table 5. 5. Figure 8 shows the inundated period for each site, however, it cannot be seen that any data show 20% or 90% for SD1 or SD5 in the subgraphs of CI7 and CI8. 6. The section of concluding remarks should be enhanced. The current conclusions are not intensive enough to show the findings of this paper. At least some quantitative analysis can be summarized and presented for readers to better understand how this work promotes the current risk assessments of compound flood hazards. 7. There are some mistakes in grammar and spelling and the authors also did not pay enough attention to punctuation, which makes this manuscript more like a draft.

---

## Referee Comment (RC2) · Anonymous Referee #2 · 18 Aug 2020

The paper describes a modelling study aimed at assessing flooding hazard at eight specific sites in the coastal areas of Connecticut as a consequence of compound flooding events, considering both current and future climate conditions.

**Major points**

1. The title is awkward to read. I suggest something as "Flood impact on coastal critical infrastructures considering compound flood events in current and future climate".

2. The Introduction is quite general and not specific enough. What the Author describes as a "a dynamic framework to project the combined hazard" is nothing else that a hydrological model and a hydrodynamic model run in cascade and

forced with both actual and synthetic data. Nonetheless, an estimation of the expected frequency is fundamental when treating compound events. This aspect is quite lacking in the paper.

Many statements are quite imprecise. For example, it is stated that the focus is on coastal power grid substations, but this is not correct. No information is given about the chance of malfunctioning of power grid substations due to flooding. Are these substations built up to tolerate a given water depth? The paper only deals with the water depths at eight locations in which power grid substations are present, which is quite another (preliminary) issue. Moreover, at the end of the Introduction, two main questions are reported. First, it is said that the present work forms the basis on which to address these two questions (which is correct), then it is said that these questions are investigated, which is incorrect.

3. Model calibration/validation. I'm not an expert of meteorological models, so I'm not commenting on. But for what concerns hydrological and hydrodynamic models, I have substantial concerns.

As for the hydrological model, the use of information on land use, land cover, and imperviousness ratio does not imply that an overparameterized model (as all spatially explicit and hyper-resolution model are) provides reliable results. The fact that the model was successfully verified in river basins within Connecticut, where all the watersheds simulated in this study reside, does not assure the model reliability in different river basins. Indeed, it is common that different rivers in the same country show very different hydrological behaviours. Calibration and validation should have been performed for the rivers considered in this study, and for the actual events (Sandy and Irene) the outcome of the model should have been compared with some measured data (no measured data within all the modelled domain seems quite an unrealistic picture).

4. It is simply unacceptable that a riverine model is set-up using LiDAR data also for

the submerged channel beds. Bed elevations MUST be corrected using proper bathymetric data (multibeam, cross sections, etc.) to obtain reliable results. Contrarily to what the Authors stated, it cannot be concluded that neglecting submerged channel bed, which results in an underestimation of channel conveyance capacity, would lead to an overestimation of the flood extent. A channel with a lower capacity can also confine an inundated area, whereas a greater conveyance capacity can cause further flooding as well. Furthermore, the model is validated considering water depth only, and not flood extent.

5. Figure 4 shows a comparison between modelled and measures water depth. Considering that two real flooding events (Sandy and Irene hurricanes) were simulated, I was expecting a comparison for these two events. Modelled water depths are reported in the figure using boxplot (instead of single values referring to these two real hurricane events), but it is not said from which set of simulations these boxplots are derived from.

Finally, I agree with the comments raised by the Reviewer 1. In general, the manuscript should be substantially revised and arranged with far greater rigour.

**Minor points**

- l. 55: "riverine models cannot capture the risk from tide-surge-SLR effects". In what a sense? While it is true that, traditionally, one looks at the river or at the coast one at a time, riverine models can naturally capture the risk induced by tide-surge-SLR on flooding in the form of higher free-surface elevations for tailwater effects, when forced with proper downstream boundary conditions. Moreover, if the riverine model includes floodable areas adjacent to the coast, the same hydrodynamic model can be used to assess coastal flooding too, it's only a matter of boundary conditions.

[Figure]

- l. 56-57: Depending of what is meant for "riverine models", "the modelling of individual flood drivers separately mischaracterizes the true risk of flooding" is not a rigorous statement, as what the Authors affirms is true only when the effects of compound events are worsen than the sum of effects due to single forcing events.

- l. 56: Barnard et al. 2017 is not present in the Bibliography.

- l. 73: "in frequency"? The sense of this sentence remains obscure to me.

- l. 90: please repeat what kind of substations.

- l. 109-111: I cannot recognize subsection a, b, and c in the text.

- l. 157: extent of what? depth of what? (water, of course).

- l. 160: How were the building footprint used in the model? So many different approaches have been proposed...

- l. 279: Please explain how cumulative distribution function (CDF) of maximum flood depths were computed.

- In the Bibliography, items are not ordered alphabetically, nor they are given the proper stylisation.

---

## Author Comment (AC1) · 8 Sep 2020

We wish to thank the reviewer for their valuable suggestions. We agree with most of the critiques raised during the review process, and we will do our best to incorporate them in the revised paper.

Here is a detailed response [in italics] to each point raised during the review [underlined font].

1. The literature review for frequency-based effect estimate of compound-event flooding (Line 69-76) is obscure. What is the missing link in the current research and why most of the studies failed to or avoided to explore the frequency and risk assessment of the compound flooding? It seems in this study the authors designed several compound scenarios to consider the probability of precipitation and surge as a solution to the shortcoming associated with compound flood risk assessment. If this is the case, more details on the related theories and methodologies should be presented in the introduction.

*We thank the reviewer for this comment. We will modify the introduction to provide a better framework for this study and highlight the importance of this work. We will improve the literature background to highlight more clearly what is missing in current research, and what this work is addressing. We will rephrase the introduction, and we will clarify better adding the text below*

*"In low-lying coastal areas, the co-occurrence of high sea level and precipitation, resulting in large runoff may cause compound flooding [CF] [Bevacqua et al., 2019]. When the two hazards interact, the resulting impact can be worse than when they occur individually. Both storm surges and heavy precipitation, as well as their interplay, are likely to change in response to global warming [Field et al., 2012].*

*Major research has been conducted on the assessment of damages to the power system components or other related infrastructures, and proposing design and operation countermeasures and remedies [i.e. Kwasinski et al. 2009; Reed et al. 2010; Abi-Sarma and Henry, 2011; Chang et al., 2007; de Bruijn et al., 2019; Pearson et al., 2018; Pant et al., 2018; Dawson, 2018]. Nonetheless, despite the CF relevance, a comprehensive hazard assessment on critical infrastructure is missing, and no studies have examined CF in the future.*

*The first step to investigate and assess the impact of CF on the power grid is to perform a systematic risk analysis. To deal with CF coming threats and challenges, there is a need to develop efficient frameworks for exploring a wide range of actual and what-if scenarios in a system that could inform short- and long-term decisions. Scientists must investigate not only how severe these events might be but also how commonly they are likely to occur. We propose a new strategy for providing this information: identify water levels and extent nearby critical infrastructure by observing real-world phenomena and drawing information from simulations.*

*When a hurricane approaches, providing a few extra hours' notice for infrastructure management is critical. By simulating the impact using possible storm paths, this framework offers more accurate medium-term risk evaluation. It can be used to assess the vulnerability of the infrastructures to current and future events."*

2. In section 2.3, it is better to use a table to describe these compound scenarios and their related hurricanes, SLR, tide conditions, and other attributes.

*We thank the reviewer for this comment. The manuscript currently presents the scenario in two different tables [table 4 and 5]. These two tables represent the peak flow, precipitation, total water level (tide + SLR) for all the scenarios at all the study locations. Considering this comment, in the revised manuscript we will provide an updated table, similar to the below one.*

*Table R1: Tide, Surge at the maximum total water level instance, Accumulated precipitation & peak flows (with return period reported within brackets) for the simulated scenarios. The readers should refer to Chapter*

*2.2 for a detailed description of each hurricane scenario (IR for Irene, SD for Sandy, FL for Florence). Events marked with "*" denotes scenarios having sea level rise (SLR) added to the surge. Critical infrastructures are labelled CI1 to CI8 according to Table 1.*

| Scenarios | | CI1 | CI2 | CI3 | CI4/ CI5 | CI6 | CI7 | CI8 |
|---|---|---|---|---|---|---|---|---|
| FL1 | Tide (m) | 0.99 | 0.99 | 0.99 | 0.94 | 0.94 | 0.94 | 0.17 |
| | Surge (m) | 2.51 | 2.51 | 2.51 | 2.56 | 2.46 | 2.56 | 3.33 |
| | Accumulated precipitation (mm) | 128.5 | 147.5 | 165.1 | 192 | 203.9 | 200.7 | 289.2 |
| | Peak flow, m3/s (return period) | 51.3(<2) | 87.4(5) | 74.9(<2) | 106.1(13) | 113.3(8) | 143.2(51) | 93.1(6) |
| FL2* | Tide (m) | 0.99 | 0.99 | 0.99 | 0.94 | 0.94 | 0.94 | 0.17 |
| | Surge (m) | 3.12 | 3.12 | 3.12 | 3.17 | 3.07 | 3.17 | 3.93 |
| | Accumulated precipitation (mm) | 128.5 | 147.5 | 165.1 | 192 | 203.9 | 200.7 | 289.2 |
| | Peak flow, m3/s (return period) | 51.3(<2) | 87.4(5) | 74.9(<2) | 106.1(13) | 113.3(8) | 143.2(51) | 93.1(6) |
| SD1 | Tide (m) | 0.82 | 0.82 | 0.82 | 0.4 | 0.4 | 0.4 | 0.01 |
| | Surge (m) | 2.37 | 2.37 | 2.37 | 2.3 | 2.3 | 2.3 | 1.87 |
| | Accumulated precipitation (mm) | 24.8 | 24.7 | 21.5 | 17 | 17.7 | 15.1 | 8.9 |
| | Peak flow, m3/s (return period) | 3.4(<2) | 9.3(<2) | 3.3 (<2) | 4.7(<2) | 1.3(<2) | 0.9(<2) | 0.03(<2) |
| SD2 | Tide (m) | 1.01 | 1.01 | 1.01 | 1.13 | 1.13 | 1.13 | -0.15 |
| | Surge (m) | 2.56 | 2.56 | 2.56 | 2.8 | 2.8 | 2.8 | 1.95 |
| | Accumulated precipitation (mm) | 24.8 | 24.7 | 21.5 | 17 | 17.7 | 15.1 | 8.9 |
| | Peak flow, m3/s (return period) | 3.4(<2) | 9.3(<2) | 3.3 (<2) | 4.7(<2) | 1.3(<2) | 0.9(<2) | 0.03(<2) |
| SD3* | Tide (m) | 1.01 | 1.01 | 1.01 | 1.13 | 1.13 | 1.13 | -0.15 |
| | Surge (m) | 3.12 | 3.12 | 3.12 | 3.4 | 3.4 | 3.4 | 2.564016 |
| | Accumulated precipitation (mm) | 24.8 | 24.7 | 21.5 | 17 | 17.7 | 15.1 | 8.9 |
| | Peak flow, m3/s (return period) | 3.4(<2) | 9.3(<2) | 3.3 (<2) | 4.7(<2) | 1.3(<2) | 0.9(<2) | 0.03(<2) |
| SD4 | Tide (m) | 1.01 | 1.01 | 1.01 | 1.13 | 1.13 | 1.13 | -0.15 |
| | Surge (m) | 2.56 | 2.56 | 2.56 | 2.8 | 2.8 | 2.8 | 1.95 |
| | Accumulated precipitation (mm) | 555.3 | 546.9 | 526.8 | 338.2 | 330.2 | 316.6 | 323.7 |
| | Peak flow, m3/s (return period) | 242.4(316) | 319.1(326) | 201.7(28) | 178.3(98) | 168.4 (48) | 197.0(301) | 94.7(6) |
| SD5* | Tide (m) | 1.01 | 1.01 | 1.01 | 1.13 | 1.13 | 1.13 | -0.15 |
| | Surge (m) | 3.12 | 3.12 | 3.12 | 3.4 | 3.4 | 3.4 | 2.564016 |
| | Accumulated precipitation (mm) | 555.3 | 546.9 | 526.8 | 338.2 | 330.2 | 316.6 | 323.7 |
| | Peak flow, m3/s (return period) | 242.4(316) | 319.1(326) | 201.7(28) | 178.3(98) | 168.4 (48) | 197.0(301) | 94.7(6) |
| IR1 | Tide (m) | 1.16 | 1.16 | 1.16 | 1.1 | 1.1 | 1.1 | 0.93 |
| | Surge (m) | 1.94 | 1.94 | 1.35 | 1.42 | 1.42 | 1.42 | 1.1 |
| | Accumulated precipitation (mm) | 187.8 | 177.8 | 173.5 | 98.1 | 91.6 | 86.1 | 58.5 |
| | Peak flow, m3/s (return period) | 158.5(56) | 201.1(58) | 126.7 (26) | 93.9(5) | 85.7(5) | 93.5(5) | 30.8(3) |
| IR2* | Tide (m) | 1.16 | 1.16 | 1.16 | 1.1 | 1.1 | 1.1 | 2 |
| | Surge (m) | 2.54 | 2.54 | 1.94 | 2.03 | 2.03 | 2.03 | 1.7 |
| | Accumulated precipitation (mm) | 187.8 | 177.8 | 173.5 | 98.1 | 91.6 | 86.1 | 58.5 |
| | Peak flow, m3/s (return period) | 158.5(56) | 201.1(58) | 126.7 (26) | 93.9(5) | 85.7(5) | 93.5(5) | 30.8(3) |

3. In section 2.4, which site does Figure 5 present for? The red rectangle shows a window of 48 hrs, not 24 hrs. What criterion is used for selecting the window size?

*We thank the reviewer for this comment. The rectangle was to bring attention to the peak, and highlight the changes in depth for the different scenarios. In the revised manuscript, we will improve the figure to avoid confusion, and we will add clarification on the caption regarding the location.*

4. In Lines such as 230, 237, Table 4 should be Table 5.

*We thank the reviewer for this comment. We will fix these mistakes in the revised manuscript.*

5. Figure 8 shows the inundated period for each site, however, it cannot be seen that any data show 20% or 90% for SD1 or SD5 in the subgraphs of CI7 and CI8.

*We thank the reviewer for this comment. If a critical infrastructure shows 0%, it means that for that scenario/event the water didn't reach the substation itself. This could be due to the water flooding other upstream locations, and therefore draining away from the station, or because the topography of the landscape prevents water from reaching the area for some specific events. We hope this can clarify this question, and we will add some comments in the manuscript about this.*

6. The section of concluding remarks should be enhanced. The current conclusions are not intensive enough to show the findings of this paper. At least some quantitative analysis can be summarized and presented for readers to better understand how this work promotes the current risk assessments of compound flood hazards.

*We thank the reviewer for this comment. We will enhance the concluding remarks. As per the reviewer's suggestion, we will highlight the main findings of the study. We will summarize the overall impact on the critical structure in terms of flood extent and depth. We will also comment strongly on how the existing guidelines should be reformulated to protect the critical infrastructures based on our findings.*

7. There are some mistakes in grammar and spelling and the authors also did not pay enough attention to punctuation, which makes this manuscript more like a draft.

*We thank the reviewer for their valuable suggestion. We will proof-read the paper and improve grammar and spelling before submitting the revised manuscript.*

*References*

A.      Kwasinski, W.W. Weaver, P.L. Chapman and P.T. Krein, "Telecommunications Power Plant Damage Assessment for Hurricane Katrina – Site Survey and Follow-Up Results," IEEE Systems Journal, vol. 3, no. 3, pp. 277–287, Nov. 2009.
B.      D.A. Reed, M.D. Powell and J.M. Westerman, "Energy Supply System Performance for Hurricane Katrina," Journal of Energy Engineering, pp. 95–102, Dec. 2010.
C.      N. Abi-Samra and W. Henry, "Actions Before and After a Flood – Substation Protection and Recovery from Weather Related Water Damage," IEEE Power & Energy Magazine, pp. 52–58, Mar/Apr. 2011.
D.      Chang, S. E., McDaniels, T. L., Mikawoz, J., & Peterson, K.: Infrastructure failure interdependencies in extreme events: power outage consequences in the 1998 Ice Storm. Natural Hazards, 41(2), 337–358. https://doi.org/10.1007/s11069-006-9039-4, 2007.
E.      Dawson, R. J., Thompson, D., Johns, D., Wood, R., Darch, G., Chapman, L., Hughes, P. N., Watson, G. V. R., Paulson, K., Bell, S., Gosling, S. N., Powrie, W. and Hall, J. W.: A systems framework for national assessment of climate risks to infrastructure, Philos. Trans. R. Soc. A Math. Phys. Eng. Sci., 376(2121), doi:10.1098/rsta.2017.0298, 2018.
F.      de Bruijn, K. M., Maran, C., Zygnerski, M., Jurado, J., Burzel, A., Jeuken, C. and Obeysekera, J.: Flood resilience of critical infrastructure: Approach and method applied to Fort Lauderdale, Florida, Water (Switzerland), 11(3), doi:10.3390/w11030517, 2019.
G.      Pant, R., Thacker, S., Hall, J. W., Alderson, D. and Barr, S.: Critical infrastructure impact assessment due to flood exposure, J. Flood Risk Manag., 11(1), 22–33, doi:10.1111/jfr3.12288, 2018.
H.      Pearson, J., Punzo, G., Mayfield, M., Brighty, G., Parsons, A., Collins, P., Jeavons, S. and Tagg, A.: Flood resilience: consolidating knowledge between and within critical infrastructure sectors, Environ. Syst. Decis., 38(3), 318–329, doi:10.1007/s10669-018-9709-2, 2018.

---

## Author Comment (AC2) · 8 Sep 2020

**Response to reviewer' comments**
On the manuscript **nhess-2020-132**
revised for publication in
NHESS

We wish to thank the reviewer for their valuable suggestions. We agree with most of the critiques raised during the review process, and we will do our best to incorporate them in the revised paper.

Here is a detailed response [in italics] to each point raised during the review [underlined font].

1. The title is awkward to read. I suggest something as "Flood impact on coastal critical infrastructures considering compound flood events in current and future climate".

*We thank the reviewer for this suggestion, and we will consider changing the title*

2. The Introduction is quite general and not specific enough. What the Author describes as a "a dynamic framework to project the combined hazard" is nothing else that a hydrological model and a hydrodynamic model run in cascade and forced with both actual and synthetic data.

*We thank the reviewer for this comment: we will modify the Introduction to provide a better framework for this study and highlight the importance of this work. We will improve the literature background to highlight in a more straightforward way what is missing in current research, and what this work is addressing. We will rephrase the Introduction, and we will clarify better adding the text below*

*"In low-lying coastal areas, the co-occurrence of high sea level and precipitation, resulting in large runoff may cause compound flooding [CF] [Bevacqua et al., 2019]. When the two hazards interact, the resulting impact can be worse than when they occur individually. Both storm surges and heavy precipitation, as well as their interplay, are likely to change in response to global warming [Field et al., 2012].*

*Major research has been conducted on the assessment of damages to the power system components or other related infrastructures, and proposing design and operation countermeasures and remedies [i.e. Kwasinski et al. 2009; Reed et al. 2010; Abi-Sarma and Henry, 2011; Chang et al., 2007; de Bruijn et al., 2019; Pearson et al., 2018; Pant et al., 2018; Dawson, 2018]. Nonetheless, despite the CF relevance, a comprehensive hazard assessment on critical infrastructure is missing, and no studies have examined CF in the future.*

*The first step to investigate and assess the impact of CF on the power grid is to perform a systematic risk analysis. To deal with CF coming threats and challenges, there is a need to develop efficient frameworks for exploring a wide range of actual and what-if scenarios in a system that could inform short- and long-term decisions. Scientists must investigate not only how severe these events might be but also how commonly they are likely to occur. We propose a new strategy for providing this information: identify water levels and extent nearby critical infrastructure by observing real-world phenomena and drawing information from simulations.*

*When a hurricane approaches, providing a few extra hours' notice for infrastructure management is critical. By simulating the impact using possible storm paths, this framework offers more accurate medium-term risk evaluation. It can be used to assess the vulnerability of the infrastructures to current and future events."*

a.      Nonetheless, an estimation of the expected frequency is fundamental when treating compound events. This aspect is quite lacking in the paper.

*We thank the reviewer for this comment. We definitely agree that a frequency estimation is critical in treating compound events. This, however, goes beyond the scope of the current manuscript.*

*For this work, our aim was to set up a modeling framework and use it to demonstrate the importance of investigating flood impacts due to compound events, based on past hurricane events and synthetic hurricane cases simulated in future climate conditions. We will address this aspect further in the discussion.*

b. Many statements are quite imprecise. For example, it is stated that the focus is on coastal power grid substations, but this is not correct.

*We thank the reviewer for this comment. Within our study sites, two are more inland [CI3 and CI4] (table 1: see hydrologic distance), nonetheless all the sites are included within the Coastal Area as defined by Connecticut General Statute (C.G.S.) 22a-94(a)*

*[https://www.cga.ct.gov/current/pub/chap_444.htm#sec_22a-94].*

*We will clarify this better in the revised manuscript and include the 'coastal area legal boundaries' in Figure 1.*

3. No information is given about the chance of malfunctioning of power grid substations due to flooding. Are these substations built up to tolerate a given water depth?

*We thank the reviewer for this comment. Due to confidentiality, we cannot provide exact information related to the critical water level for each infrastructure. The presented water depths are indicative numbers, useful to provide a comparison between the various events. In the revised manuscript we will give few comments about this.*

*4.* The paper only deals with the water depths at eight locations in which power grid substations are present, which is quite another (preliminary) issue. Moreover, at the end of the Introduction, two main questions are reported. First, it is said that the present work forms the basis on which to address these two questions (which is correct), then it is said that these questions are investigated, which is incorrect.

*We thank the reviewer for this comment. We want to underline that indeed, the aim of the paper was characterizing the risk for the critical infrastructures, hence why we described the water depth at the location. We, however, also investigated the water depths in the whole domain, through the CDFs, and compared the water extent to FEMA maps, to provide an overall hazard assessment. Considering the questions in the Introduction, we will rephrase this chapter, providing a more explicit description.*

5. Model calibration/validation. I'm not an expert of meteorological models, so I'm not commenting on. But for what concerns hydrological and hydrodynamic models, I have substantial concerns.

a. As for the hydrological model, the use of information on land use, land cover, and imperviousness ratio does not imply that an overparameterized model (as all spatially explicit and hyper-resolution model are) provides reliable results. The fact that the model was successfully verified in river basins within Connecticut, where all the watersheds simulated in this study reside, does not assure the model reliability in different river basins. Indeed, it is common that different rivers in the same country show very different hydrological behaviours. Calibration and validation should have been performed for the rivers considered in this study, and for the actual events (Sandy and Irene) the outcome of the model should have been compared with some measured data (no measured data within all the modelled domain seems quite an unrealistic picture).

*We thank the reviewer for this comment. We would like to clarify that the hydrologic model was calibrated and validated for the whole Connecticut river basin in the work by Shen and Anagnostou (2017). In the paper, the model was tested for Thompsonville (gauge No. 01184000) with a NSCE of 0.63. Recently, we further validated the model considering hourly flows in Housatonic River and Naugatuck River with a NSCE of 0.69. We will add this part in the manuscript, to clarify the performance of the hydrologic model.*

b. It is simply unacceptable that a riverine model is set-up using LiDAR data also for the submerged channel beds. Bed elevations MUST be corrected using proper bathymetric data (multibeam, cross sections, etc.) to obtain reliable results. Contrarily to what the Authors stated, it cannot be concluded that neglecting submerged channel bed, which results in an underestimation of channel conveyance capacity, would lead to an overestimation of the flood extent. A channel with a lower capacity can also confine an inundated area, whereas a greater conveyance capacity can cause further flooding as well. Furthermore, the model is validated considering water depth only, and not flood extent.

*We fully agree with the reviewer on the importance of bathymetry in flood inundation modeling. Unfortunately, for the investigated case studies, we do not have any information about the bathymetry of the rivers.*

*In general, the impact of inclusion/exclusion of bathymetry data on the model results will vary in its magnitude as a function of river size and flood magnitude [Cook & Merwade 2009]. For larger events in these coastal locations, flood risk is mainly dominated by defence overflow and defence breaching. As a consequence, we did not represent the flow of water in the main channel. Rather, boundary conditions were given as time series of water surface elevation imposed along the defence crests. This means that we do not require detailed bathymetric information in the upstream main channel, thereby considerably simplifying the modelling problem (as also suggested in Bates et al. 2013). As well, the model parameters were calibrated to obtain realistic water depths, as compared to the High-water marks in selected locations, and this allowed us to obtain realistic simulations for Sandy.*

*Considering the reviewer comments, we will include further clarification about the model validation.*

*As an example, the following paragraphs illustrate how the proposed model, for such extreme events, provides realistic simulations, even when compared to running the model accounting for bathymetry.*

*Given the lack of bathymetry data for the case studies, as an example, we applied a Discharge Correction Technique (DCT) to the hydrologically simulated discharge. DCT is based on the assumption that a given flow discharge can be separated in two components: the bankfull discharge, below the assessed water surface, and the discharge exceeding the LiDAR discharge, above the assessed water surface (Bradbrook et al. 2004). To evaluate the bankfull discharge, we considered regional curves (Ahearn, 2004).*

*Fig R1 shows  for hurricane Irene [actual event], a comparison between the CREST-simulated discharge, and the DCT one. The results of the simulation carried out as presented in the manuscript, VS the simulation corrected using the DCT for CI1 is shown in Fig R2.*

[Figure]

*Figure R1: example of DCT as compared to CREST simulated discharge*

[Figure]

[Figure]

*Figure R2: Maximum flood depth during the actual Irene event: left: streamflow with DCT, right: Streamflow without DCT, as presented in the submitted manuscript.*

*Regarding the model validation, see our response to the below point*

6. Figure 4 shows a comparison between modelled and measured water depth. Considering that two real flooding events (Sandy and Irene hurricanes) were simulated, I was expecting a comparison for these two events. Modelled water depths are reported in the figure using boxplot (instead of single values referring to these two real hurricane events), but it is not said from which set of simulations these boxplots are derived from.

*We thank the reviewer for this comment. As validation data, we only have information for Sandy, not for Irene. Hence we based our comparison to that event. In the revised manuscript will clarify better figure 4: to allow for comparison, we evaluate water depth within a 10x10m radius around the high water marks, to avoid issues due to the presence of buildings in the DTM; hence we represented the figure using boxplots.*

*Regarding the validation of the flood extent, we will provide further assessment in the revised manuscript.*

*As for the water depth, the most accurate available information for flood extent is only available for Sandy. For this event, CTEco (FEMA,CT DEEP, 2013) provides a map of the storm surge, created from field-verified High Water Marks and Storm Surge Sensor data from the USGS. For Connecticut, the vertical value is water depth above the ground in feet. For comparison purposes, we here provide a visual quality assessment of our model (Fig.R3 a, c), as compared to these maps (Fig. R3 d,e), for two selected locations (CI1 and CI2).*

[Figure]

Service Layer Credits: Source: USGS, EPA
FEMA, CT DEEP

*Figure R3: comparison between the results of the proposed model for two selected locations (a,c, CI1 and CI2 respectively) and the maximum surge extent as proposed by CtEco (c,d respectively).*

*Considering the reviewer comments, we will improve the discussion of the model validation in the revised manuscript.*

7. Finally, I agree with the comments raised by the Reviewer 2. In general, the manuscript should be substantially revised and arranged with far greater rigor.

*We thank the reviewer for their valuable suggestion. We will proof-read the paper and improve grammar and spelling before resubmitting the revised manuscript.*

**Minor points**

• l. 55: "riverine models cannot capture the risk from tide-surge-SLR effects". In what a sense? While it is true that, traditionally, one looks at the river or at the coast one at a time, riverine models can naturally capture the risk induced by tidesurge-SLR on flooding in the form of higher free-surface elevations for tailwater effects, when forced with proper downstream boundary conditions. Moreover, if the riverine model includes floodable areas adjacent to the coast, the same hydrodynamic model can be used to assess coastal flooding too, it's only a matter of boundary conditions.

*We thank the reviewer for this comment. We will deeply rephrase the Introduction and clarify this part better, explaining the importance of correct setting the downstream boundary conditions.*

l. 56-57: Depending of what is meant for "riverine models", "the modelling of individual flood drivers separately mischaracterizes the true risk of flooding" is not a rigorous statement, as what the Authors affirms is true only when the effects of compound events are worsen than the sum of effects due to single forcing events

*We will rephrase this sentence as follows "The modeling of individual flood drivers separately might mischaracterize the true risk of flooding, especially when the effects of compound events are worse than the sum of effects due to single forcing events."*

• l. 56: Barnard et al. 2017 is not present in the Bibliography.

*We will double-check all the references and fix them*

• l. 73: "in frequency"? The sense of this sentence remains obscure to me.

*We will rephrase this sentence. "Some authors have characterized the frequency of compound flooding, and provide approaches to risk assessment based on the joint probability of precipitation and surge (Bevacqua et al., 2019; Wahl et al., 2015)."*

• l. 90: please repeat what kind of substations.

*We will fix this*

• l. 109-111: I cannot recognize subsection a, b, and c in the text.

*We will fix this*

• l. 157: extent of what? depth of what? (water, of course).

*We will rephrase and be more precise*

• l. 160: How were the building footprint used in the model? So many different approaches have been proposed. . .

*We thank the reviewer for this comment. In the manuscript we will explain more clearly how we approached this.*

*For the simulations we considered a DTM [bare ground elevation]. To better represent the impacts of urban establishments on inundation dynamics, solid urban features such as houses and buildings which obstruct flow of storm water were added to the bare-earth DTM. To this purpose, we considered the building footprints from CtECO, 2012 and identified positions of buildings and houses in the DEM by increasing the elevation of the pixels inside of the building footprint polygons by an arbitrary height of 4.5 m ~ 15 ft, assuming one-story buildings.*

• l. 279: Please explain how cumulative distribution function (CDF) of maximum flood depths were computed.

*We computed a Cumulative Distribution Function that describes the probability that a particular value for a random variable will be exceeded. We did this using all the depth values of all the grid of the simulation domain, for the time step when the inundation was maximum. We have treated the depth values as random variables and used the existing function "cdf" in MATLAB to plot the CDF curves.*

• In the Bibliography, items are not ordered alphabetically, nor they are given the proper stylisation.

*We will double-check all the references and fix them in the revised manuscript*

*References:*

A.     *Shen, X., Anagnostou, E. N. (2017),A Framework to Improve Hyper-Resolution Hydrologic Simulation in Snow-Affected Regions. Journal of Hydrology, 552, 1–12. https://doi.org/10.1016/j.jhydrol.2017.05.048, 2017*

B.      *Cook A., Merwade, V., (2009) Effect of topographic data, geometric configuration and modeling approach on flood inundation mapping. Journal of Hydrology, 377, 1–2, 20, 131-142* *https://doi.org/10.1016/j.jhydrol.2009.08.015*

C.      *Bradbrook, K., Lane, S.N., Waller, S.G., Bates, P.D. (2004).Two dimensional diffusion wave modelling of flood inundation using a simplified channel representation. Int. J. River Basin Manag. 2, 211– 223*

D.      *Bates, P. D., Dawson, R. J., Hall, J.W., Horritt, M.S., Nicholls, R. J., Wicks, J., Hassan, M. A. A. M. (2005). Simplified two-dimensional numerical modelling of coastal flooding and example applications. Coastal Engineering, 52, 793–810.*

E.      *Ahearn E.A., (2004). Scientific Investigations Report 2004-5160, https://doi.org/10.3133/sir20045160*

F.      *FEMA,      CT      DEEP      (2013).      Coastal      Hazards      Map      Viewer      Information* *http://www.cteco.uconn.edu/viewers/coastalhazards.htm#surge*

---

## Author Response (AR1)

**Response to reviewer' comments**
On the manuscript **nhess-2020-132**
revised for publication in
NHESS

We wish to thank the reviewer for their valuable suggestions. We agree with most of the critiques raised during the review process, and we did our best to incorporate them in the revised paper.

Considering the reviewers' suggestions, we deeply modified the introduction and the discussion of the paper. New figures have been added to discuss further the validation of the proposed models [i.e. Fig. 5 in the revised manuscript], and some figures and their captions have been modified to improve clarity. We also combined the information of tables 4 and 5 to a new table (now table 3). We added a new discussion on the correlation between differences with FEMA extents, and the provided simulations (Also added in table 4, in the current submission), and we re-structured the discussion and the conclusion.

Here is a detailed response [in italics] to each point raised during the review [underlined font].

**Response to reviewer #1:**

1. The literature review for frequency-based effect estimate of compound-event flooding (Line 69-76) is obscure. What is the missing link in the current research and why most of the studies failed to or avoided to explore the frequency and risk assessment of the compound flooding? It seems in this study the authors designed several compound scenarios to consider the probability of precipitation and surge as a solution to the shortcoming associated with compound flood risk assessment. If this is the case, more details on the related theories and methodologies should be presented in the introduction.

*We thank the reviewer for this comment. We modified the introduction to provide a better framework for this study and highlighted the importance of this work. We improved the literature background to highlight more clearly what is missing in current research, and what this work is addressing. We rephrased and reorganized the introduction. Some key changes are as follows:*

*Line 36- 39: Concurrent with the rise in event intensities, the elevated damage, and disruption caused by compound flooding (CF) to critical infrastructure (CI) and services, including electrical systems, water, and sewage treatment facilities, and other utilities that underpin modern society, have substantial adverse socioeconomic impacts, especially in low-lying coastal areas, where almost 40 percent of people in the United States live (NOAA, 2013).*

*Line 45- 54: Recent studies have underlined the importance of understanding and quantifying the flood impacts on critical infrastructure, and their broader implications in risk management and catchment-level planning (Chang et al., 2007; McEvoy et al., 2012; Ziervogel et al., 2014; de Bruijn et al., 2019; Pearson et al., 2018; Pant et al., 2018; Dawson, 2018). Some authors have estimated the frequency of compound flooding and provide approaches to risk assessment based on the joint probability of precipitation and surge (Bevacqua et al., 2019; Wahl et al., 2015). The spatial extent and depth of compound flooding can vary in frequency (Quinn, et al., 2019) if any of the components of CF is not taken into consideration while evaluating flood frequency. Both storm surges and heavy precipitation, and their interplay, are likely to change in the future (Field et al., 2012, Dottori et al., 2018; Blöschl et al., 2017; Muis et al., 2016; Marsooli et al., 2019; Vousdoukas et al., 2018). Nonetheless, the effects of CF, considering the climate change impact, have not been thoroughly explored yet.*

*Line 56- 69: We present a hydrologic-hydrodynamic modeling framework to evaluate the integrated impact of flood drivers causing CF by synthesizing current and future scenarios. This study enables the quantitative measurement of CF hazard cast on critical infrastructures in terms of flood depth and flood extent by*

*observing actual storm-induced floods and drawing information from synthetic scenarios. To project the combined flood hazard in future climate conditions, we integrated the effects of SLR, tides, and synthetic hurricane event simulations into the flood hazard exposure.*

*Even though past research on the assessment of damages to the power system components or other related infrastructures has proposed design and operation countermeasures and remedies (i.e. Kwasinski et al. 2009; Reed et al. 2010; Abi-Sarma and Henry, 2011; Chang et al., 2007; de Bruijn et al., 2019; Pearson et al., 2018; Pant et al., 2018; Dawson, 2018), these studies lack a comprehensive hazard assessment on power grid components, and.potential changes due to climate change.*

2. In section 2.3, it is better to use a table to describe these compound scenarios and their related hurricanes, SLR, tide conditions, and other attributes.

*We thank the reviewer for this comment. We combined the information of tables 4 and 5 to table 3 in the submitted manuscript. We have rephrased section 2.3 accordingly.*

3. In section 2.4, which site does Figure 5 present for? The red rectangle shows a window of 48 hrs, not 24 hrs. What criterion is used for selecting the window size?

*We thank the reviewer for this comment. The rectangle was simply to bring attention to the peak and highlight the changes in depth for the different scenarios. We have removed the rectangle from the figure and clarified more in the text. The figure number is now "Figure 6". The site information is added to the caption of the figure.*

[Figure]

*Figure 6: Example of time series of depth values for the different scenarios of Sandy event at CI3 [SD1 to SD5, readers should refer to Table 3 and chapter 2.4 for specification on the scenarios]*

4. In Lines such as 230, 237, Table 4 should be Table 5.

*We thank the reviewer for this comment. We have fixed the text according to the modified table and figure numbers.*

5. Figure 8 shows the inundated period for each site; however, it cannot be seen that any data show 20% or 90% for SD1 or SD5 in the subgraphs of CI7 and CI8.

*We thank the reviewer for this comment. We clarified section 3.3. More in detail, we added the following paragraph.*

*line 368-371: ……. If a critical infrastructure shows 0%, it means that for that scenario/event the water didn't reach the substation at all, at least during the simulated timeframe. This could be due to the water flooding other upstream locations, and therefore draining away from the station, or because the topography of the landscape actually prevented water from reaching the area for some specific events.*

6. The section of concluding remarks should be enhanced. The current conclusions are not intensive enough to show the findings of this paper. At least some quantitative analysis can be summarized and presented for readers to better understand how this work promotes the current risk assessments of compound flood hazards.

*We thank the reviewer for this comment. We rephrased the concluding remarks as per the reviewer's suggestion.*

7. There are some mistakes in grammar and spelling and the authors also did not pay enough attention to punctuation, which makes this manuscript more like a draft.

*We thank the reviewer for their valuable suggestion. We proofread for grammar, spellings, and punctuations for better quality and readability.*

   a. Nonetheless, an estimation of the expected frequency is fundamental when treating compound events. This aspect is quite lacking in the paper.

*We thank the reviewer for this comment. We definitely agree that a frequency estimation is critical in treating compound events. This, however, goes beyond the scope of the current manuscript. For this work, we aimed to set up a modeling framework, and use it to demonstrate the importance of compound events on infrastructure flooding based on past hurricane events and synthetic hurricane cases simulated in future climate conditions. In the discussion we have stated our intention for future works with the following text:*

*Line 449- 454: Future research should consider improved estimation methods, including more detailed information on the variability of river properties (i.e. depth and width). Future works should also relate the frequency of inundation depths to return periods of precipitation, river flows, and surges, as well as differentiate among the individual effects of the components to determine the role of each in flooding impact. This can be a very useful piece of information for deciding whether and where to take measures in terms of flood occurrence and the potential relocation of CI to avoid catastrophic compound flood events.*

Many statements are quite imprecise. For example, it is stated that the focus is on coastal power grid substations, but this is not correct.

*We thank the reviewer for this comment. We have added the following text to address this comment:*

*Line 89- 91: Among the case study sites, two CIs are relatively inland [CI3 and CI4] (table 1: see hydrologic distance. Figure 1: see coastal boundary), nonetheless all the sites are included within the Coastal Area as defined by Connecticut General Statute (CGS) 22a-94(a) [https://www.cga.ct.gov/current/pub/chap_444.htm#sec_22a-94].*

*We also included the boundary in Figure 1 (see below).*

[Figure]

**Figure 1: Study area with associated watersheds and simulation domains. Locations of substations and USGS high water marks are also shown. Red circles in the top left-hand panel, and marked with A, B, and C are highlighted in panels A to C respectively. Background map by ESRI web-services, provided by UConn/CTDEEP, Esri, Garmin, USGS, NGA, EPA, USDA, NPS**

3. No information is given about the chance of malfunctioning of power grid substations due to flooding. Are these substations built up to tolerate a given water depth?

*We thank the reviewer for this comment. Due to confidentiality, we cannot provide exact information related to the critical water level for each infrastructure. The presented water depths are indicative numbers, useful to provide a comparison between the various events. In section 2.4, lines 253-260, we discussed the threshold used for the analysis of the flood depth at each station.*

*4.* The paper only deals with the water depths at eight locations in which power grid substations are present, which is quite another (preliminary) issue. Moreover, at the end of the Introduction, two main questions are reported. First, it is said that the present work forms the basis on which to address these two questions (which is correct), then it is said that these questions are investigated, which is incorrect.

*We thank the reviewer for this comment. We would like to underline that indeed, the paper aims to characterize the risk for critical electric grid infrastructures, and this is why we analyzed the water depth at the selected locations. We, however, also investigated the water depths in the entire model domain, by presenting the CDFs, and comparing the water extent to the FEMA 100 year flood maps, which provides an overall hazard assessment of the studied compound events. In the revised manuscript we rephrase the questions in the introduction, to provide a clearer description of the focus of our study. The new questions are:*

*Line 70- 74: The scenario-based analysis of this study formed the basis on which to address two questions:*

*(1) What are the characteristics of the tropical storm-related inundation, considering the compound effect of riverine and coastal flooding coinciding or not with peak high tides*

*(2) Will future climate (including SLR and intensification of storms due to warmer sea surface temperatures) bring a significant increase in flood impact for the power-grid coastal infrastructures?*

5. Model calibration/validation. I'm not an expert of meteorological models, so I'm not commenting on. But for what concerns hydrological and hydrodynamic models, I have substantial concerns.

   a. As for the hydrological model, the use of information on land use, land cover, and imperviousness ratio does not imply that an overparameterized model (as all spatially explicit and hyper-resolution model are) provides reliable results. The fact that the model was successfully verified in river basins within Connecticut, where all the watersheds simulated in this study reside, does not assure the model reliability in different river basins. Indeed, it is common that different rivers in the same country show very different hydrological behaviours. Calibration and validation should have been performed for the rivers considered in this study, and for the actual events (Sandy and Irene) the outcome of the model should have been compared with some measured data (no measured data within all the modelled domain seems quite an unrealistic picture).

   *We thank the reviewer for this comment. We clarified the calibration and validation process using the following text:*

*Line 147- 151: CREST-SVAS was calibrated and validated for the whole Connecticut river basin [that contains all the investigated sites] with an NSCE of 0.63 (Shen and Anagnostou, 2017). We further validated the model considering hourly flows in two locations within the Housatonic River and Naugatuck River watersheds with*

*an NSCE of 0.69 (Hardesty et al., 2018). The quality measures indicate a satisfactory model performance at the watershed scale over the topographic region that collectively include our study sites.*

b. It is simply unacceptable that a riverine model is set-up using LiDAR data also for the submerged channel beds. Bed elevations MUST be corrected using proper bathymetric data (multibeam, cross sections, etc.) to obtain reliable results. Contrarily to what the Authors stated, it cannot be concluded that neglecting submerged channel bed, which results in an underestimation of channel conveyance capacity, would lead to an overestimation of the flood extent. A channel with a lower capacity can also confine an inundated area, whereas a greater conveyance capacity can cause further flooding as well. Furthermore, the model is validated considering water depth only, and not flood extent.

*We agree with the reviewer on the importance of bathymetry in flood inundation modeling.*

*As an example, the following paragraphs illustrate how the proposed model provides good simulations, even when compared to running the model accounting for bathymetry.*

*For this, we applied a Discharge Correction Technique (DCT) to the hydrologically simulated discharge. DCT is based on the assumption that a given flow discharge can be separated into two components: the bankfull discharge, below the assessed water surface, and the discharge exceeding the LiDAR discharge, above the assessed water surface. This technique is used commonly to assess the discharge of a compound channel and is also known as the horizontally divided channel method (Bradbrook et al. 2004). To evaluate the bankfull discharge, we considered regional curves (Ahearn, 2004). Fig R1 shows for hurricane Irene [actual event], a comparison between the CREST-simulated discharge, and the DCT one. The results of the simulation carried out as presented in the manuscript, VS the simulation corrected using the DCT for CI1 is shown in Fig R2.*

[Figure]

*Figure R1: example of DCT as compared to CREST simulated discharge*

[Figure]

[Figure]

*Figure R2: Maximum flood depth during the actual Irene event: left: streamflow with DCT, right: Streamflow without DCT, as presented in the submitted manuscript.*

*These results highlight how, for a larger event, where the floodplains are near fully flooded, the inclusion of bathymetry does not make substantial differences for the extent of flooding or the water depth. We did not include the above-explained analysis to the manuscript but based on the findings we added the following text to the revised manuscript:*

*Line 163- 169: The considered locations have no bathymetric (underwater topography) data represented in the DEM. In general, the impact of inclusion/exclusion of bathymetry data on the hydrodynamic model simulations will vary according to the river size and event severity (Cook & Merwade 2009). For the investigated events in this study, defence overflow and defence breaching mainly dominate flood risk. This means that we do not require detailed bathymetric information in the upstream main channel, thereby considerably simplifying the modeling problem (Bates et al. 2013). Therefore, we did not represent the flow of water in the main channel. Rather boundary conditions were given as time series of water surface elevation imposed along the defence crests.*

*Considering the reviewer comments, we included further clarification about the model validation, see our response to the below point*

6. Figure 4 shows a comparison between modelled and measured water depth. Considering that two real flooding events (Sandy and Irene hurricanes) were simulated, I was expecting a comparison for these two events. Modelled water depths are reported in the figure using boxplot (instead of single values referring to these two real hurricane events), but it is not said from which set of simulations these boxplots are derived from.

*We thank the reviewer for this comment. As validation data, we only have High water marks (HWM) and surge extent information for Sandy, not for Irene. Hence, we based our comparison on that event. We clarified the analysis with the following text:*

*Line 188- 191: An HWM does not necessarily indicate the maximum flood depth; rather, it can be a mark from a lower depth that lasts long enough to leave a trail. Based on this understanding, we compared the HWMs against the simulated flood depths within a 10x10m radius around the high water marks, also to avoid issues due to the presence of buildings in the DEM (Boxplots in Fig. 4).*

*Regarding the validation of the flood extent, we will provide further assessment in the revised manuscript. As for the water depth, the most accurate available information for flood extent is only available for Sandy.*

*Line 194- 199: Figure 5 shows a visual comparison for CI1 and CI2 between the simulated inundation (Fig.5 a, c), and the reference extent (Fig. 5 d,e). A slight overestimation of the flood level, ranging between 0.2 and 0.4 m, with a precision of 0.2 m or less, is observed for the inundation depths at the displayed locations, which is consistent with the results obtained locally, at the HWM locations (Fig. 4). Taking into consideration the accuracy of the inundation depth, the declared DEM accuracy (vertical RMSE ~0.3m), and the simplified modeling problem concerning bathymetry, the accuracy of the flood extent assessment was judged satisfactory.*

[Figure]

*Figure 5: Comparison between the results of the proposed model for two selected locations (a,c, CI1, and CI2 respectively) and the maximum surge extent as proposed by CtEco (c,d respectively).*

7. Finally, I agree with the comments raised by the Reviewer 2. In general, the manuscript should be substantially revised and arranged with far greater rigor.

*We thank the reviewer for their valuable suggestion. We proofread for grammar, spellings, and punctuations for better quality and readability.*

**Minor points**

• l. 55: "riverine models cannot capture the risk from tide-surge-SLR effects". In what a sense? While it is true that, traditionally, one looks at the river or at the coast one at a time, riverine models can naturally capture the risk induced by tidesurge-SLR on flooding in the form of higher free-surface elevations for tailwater effects, when forced with proper downstream boundary conditions. Moreover, if the riverine model includes floodable areas adjacent to the coast, the same hydrodynamic model can be used to assess coastal flooding too, it's only a matter of boundary conditions.

*We thank the reviewer for this comment. We have removed this part of the text and rephrased most of the introduction. Please refer to the Chapter1: Introduction.*

l. 56-57: Depending of what is meant for "riverine models", "the modelling of individual flood drivers separately mischaracterizes the true risk of flooding" is not a rigorous statement, as what the Authors affirms is true only when the effects of compound events are worsen than the sum of effects due to single forcing events

*We thank the reviewer for this comment. We have removed this part of the text and rephrased most of the introduction. Please refer to the Chapter1: Introduction.*

• l. 56: Barnard et al. 2017 is not present in the Bibliography.

*We fixed the bibliography*

• l. 73: "in frequency"? The sense of this sentence remains obscure to me.

*We rephrased the sentence to -*

*Line 48- 49: Some authors have estimated the frequency of compound flooding and provide approaches to risk assessment based on the joint probability of precipitation and surge (Bevacqua et al., 2019; Wahl et al., 2015).*

• l. 90: please repeat what kind of substations.

*We used "power grid substations" instead of "substations"*

• l. 109-111: I cannot recognize subsection a, b, and c in the text.

*We have fixed them*

• l. 157: extent of what? depth of what? (water, of course).

*We rephrased it to "extent and the maximum depth of the flood" in the revised manuscript line 153*

• l. 160: How were the building footprint used in the model? So many different approaches have been proposed. . .

*We thank the reviewer for this comment. In the manuscript we explained more clearly how we approached this with the following text:*

*Line 157-162: The inundation maps are derived using a 1m LIDAR DEM (CtECO 2016) taken as base maps for the study reaches. To better represent the impacts of urban establishments on inundation dynamics, solid urban features such as houses and buildings, which obstruct the flow of stormwater, were added to the bare-earth DEM. For this, we considered the building footprints from (CtECO, 2012) and identified positions of buildings and houses in the DEM by increasing the elevation of the pixels within the building footprint polygons by an arbitrary height of 4.5 m, assuming one-story buildings.*

• l. 279: Please explain how cumulative distribution function (CDF) of maximum flood depths were computed.

*Line 244 252: To evaluate the flood hazard in terms of flood depth, we computed a Cumulative Distribution Function (CDF) to shows the probability that the flood depth will attain a value less than or equal to each measured value. We estimated the CDF using all the depth values of all the grid of the simulation domain, for the time step when the inundation was maximum. We evaluated the depth empirical exceedance probability (Hanman et al., 2016; Lin et al., 2016; Warner and Tissot 2012) within the whole domain, considering the maximum depth at each pixel, as suggested in (Pasquier et al. 2019, Hamman et al. 2016). The benefits of this empirical approach are that it overcomes sensitivity to the choice of the distribution and does not require a definition of the distribution parameters. By comparing the empirical distributions, we can investigate how changes in the scenario characteristics modify the frequency of the maximum inundation depths.*

• In the Bibliography, items are not ordered alphabetically, nor they are given the proper stylisation.

*We have fixed the bibliography for style and missing ones. We also sorted them alphabetically.*

*References:*

[revised manuscript text omitted]

Deleted Cells
Deleted Cells
Deleted Cells
Inserted Cells
Inserted Cells
Inserted Cells
Inserted Cells

Deleted Cells
Inserted Cells
Inserted Cells

**Table 54: Overall extent of the inundated area (in km², and the relative difference (% change in parenthesis) compared to the FEMA 100yr Flood Zone and dCorr (correlation between differences in flood extent as compared by FEMA, and flow and surge peak)**

| CIs | FL1 | FL2 | SD1 | SD2 | SD3 | SD4 | SD5 | IR1 | IR2 | dCorr surge | dCorr flow |
|---|---|---|---|---|---|---|---|---|---|---|---|
| CI1 | 1.6 (-8.5) | 1.8 (2.9) | 0.9 (-48.1) | 1.4 (-21.7) | 1.9 (8.3) | 1.7 (-2.8) | 2.0 (13.9) | 1.3 (-27.5) | 1.5 (-15.9) | 0.86 | 0.40 |
| CI2 | 3.9 (134.2) | 4.0 (139.4) | 1.9 (-12.7) | 2.1 (25.6) | 2.3 (36.3) | 3.7 (123.7) | 4.8 (185.2) | 1.6 (-1.9) | 4.9 (192.2) | 0.53 | 0.55 |
| CI3 | 4.7 (2.6) | 4.9 (7.5) | 3.5 (-24.5) | 4.0 (-10.5) | 4.3 (-6.2) | 5.4 (17.5) | 7.1 (56.2) | 3.2 (-29.3) | 4.0 (-12.1) | 0.67 | 0.70 |
| CI4/CI5 | 2.7 (-8.3) | 3.2 (8.4) | 2.4 (-18.5) | 2.6 (0.3) | 3.4 (13.8) | 2.9 (2.5) | 3.6 (22.2) | 2.0 (-32.3) | 2.4 (-17.3) | 0.98 | 0.43 |
| CI6 | 0.9 (3.7) | 0.9 (13.1) | 0.7 (-14.9) | 0.8 (-10.3) | 1.0 (16.6) | 0.9 (11.4) | 1.0 (16.5) | 0.7 (-20.4) | 0.8 (-4.8) | 0.84 | 0.56 |
| CI7 | 2.5 (1.0) | 2.7 (12.5) | 1.6 (-33.9) | 2.0 (-12.8) | 2.6 (8.5) | 2.1 (-10.7) | 2.6 (7.3) | 1.9 (-23.5) | 2.3 (-7.5) | 0.81 | 0.46 |
| CI8 | 3.1 (4.5) | 3.5 (18.4) | 0.4 (-87.8) | 2.1 (-28.8) | 2.6 (-11.1) | 2.2 (-22.3) | 2.7 (-8.9) | 1.1 (-63.1) | 1.8 (-37.9) | 0.88 | 0.67 |

Note: (-) Area inundated less than FEMA's 100yr zone

Inserted Cells

Inserted Cells

---

## Referee Report (RR1)

The revised paper is certainly much improved compared with the previous versions. Some other issues must be fixed and some parts improved before publication in NHESS.

Specific points

1) When describing the hydrodynamic model setting, the Author state that "For the investigated events in this study flood risk is mainly dominated by defence overflow and defence breaching". While defence overflow can be easily computed as a function of the hydraulic head acting above the defence, defence breaching requires some model to simulate (or to account for) the breaching process and the ensuing much larger outflow (Dazzi et al., 2020; Viero et al., 2013). Any further detail on defence breaching is lacking in the paper.

2) When describing the hydrodynamic model setting, at lines 168-169 I read "we did not represent the flow of water in the main channel. Rather boundary conditions were given as time series of water surface elevation imposed along the defence crests". These sentences are not clear at all. How was the upstream boundary condition (inflow discharge hydrographs from the hydrological model) used in hydrodynamic modelling? What is intended for "main channel"? Does the second sentence refer to downstream boundary condition only? (at lines 220-221 I read "simulated peak flow used as an upstream boundary condition in HEC-RAS" that states that the flow of water is somewhere represented).

3) The "Concluding remarks" has been enlarged, rather than improved. Now conclusions are long to read, contain repetitions and, finally, are unable to convey clear messages.
   Each single paragraph is a collection of very different arguments, and concluding remarks on the same topic are dissected in different paragraphs. Please revise the structure of this last section.

Minor Points
- L. 67: Abi-Samra, not Abi-Sarma.
- L. 108: Skamarock, not Shamarock.
- L. 128: Meehl et al., 2007, not 2017.
- L. 166: Please consider adding a reference to Viero et al. (2019), as a relevant example of flooding dominated by defence overflow and defence breaching.
- L. 167: Bates e al. (2013) is referenced in the text but, in the Bibliography, I can only find Bates et al. (2005). This item (line 480) is missing the title and is not properly formatted.
- L. 258: check the reference to Figure 7 (maybe Figure 9 is the correct one).
- L. 275: Xian et al. (2015), not (2005).
- L. 472: in the item Ahearn (2004) the report title is missing.
- l. 476: Barnard et al (2017) is not referenced in the text.
- l. 492: Bradbrook et al. (2004) is not referenced in the text. Moreover, the link provided is not the official one, please change it to https://doi.org/10.1080/15715124.2004.9635233.
- L. 510: the year at the end.
- L. 514: Surname of the Authors first.
- L. 517: Danielson and Gesch is dated 2011, not 2016.
- L. 546: reference to Gerald et al. (2007), not cited in the text, is a duplicate of Meehl et al. (2007), and should be removed.
- L. 553-558: the reference to Hamman et al. (2016) is duplicated.
- L. 605: O'Donnel (2020) is not referenced in the text.

- The two references to Schumann et al., 2007 should be denoted with 2007a and 2007b.
- I found several references to a U.S.S Geological Survey throughout the Bibliography. It should read USGS, isn't it? (example l. 646)
- L. 648-653: the reference to Vousdoukas et al. (2018) is duplicated.
- L. 657: Wahl et al. (2018) is not referenced in the text.
- L. 671-676: the reference to Ziervogel et al. (2014) is duplicated.
- Please note that in the Copernicus template there is a Bibliography style aimed at formatting the Bibliography with proper (and reader-friendly) indentation.
- Figure 3 and throughout the text: put the superscript 3 in "m³/s".
- Figure 3: "Stream flow at upstream" => "Upstream boundary condition"; "Total Water Level at Downstream" => "Downstream boundary condition"
- Caption of Figure 3: "firhg-hand panel" should read "right-hand panel".

Additional references

Dazzi, S., Vacondio, R., & Mignosa, P. (2019). Integration of a Levee Breach Erosion Model in a GPU-Accelerated 2D Shallow Water Equations Code. *Water Resources Research*, *55*(1), 682-702. https://doi.org/10.1029/2018WR023826.

Viero, D. P., D'Alpaos, A., Carniello, L., & Defina, A. (2013). Mathematical modeling of flooding due to river bank failure. *Advances in Water Resources*, *59*, 82-94. https://doi.org/10.1016/j.advwatres.2013.05.011.

Viero, D. P., Roder, G., Matticchio, B., Defina, A., & Tarolli, P. (2019). Floods, landscape modifications and population dynamics in anthropogenic coastal lowlands: The Polesine (northern Italy) case study. *Science of The Total Environment*, *651*, 1435-1450. https://doi.org/10.1016/j.scitotenv.2018.09.121.

---

## Author Response (AR2)

**Response to reviewer' comments**
On the manuscript **nhess-2020-132**
revised for publication in
NHESS

First, we wish to thank the reviewer for their valuable insights. below our *response (in italics)* to the reviewer's raised points.

1) When describing the hydrodynamic model setting, the Author state that "For the investigated events in this study flood risk is mainly dominated by defence overflow and defence breaching". While defence overflow can be easily computed as a function of the hydraulic head acting above the defence, defence breaching requires some model to simulate (or to account for) the breaching process and the ensuing much larger outflow (Dazzi et al., 2020; Viero et al., 2013). Any further detail on defence breaching is lacking in the paper.

*We thank the reviewer for this comment. For the investigated cases there was no defence breaching, the sentence was referring to the general risk in the coastal area. We clarified the text, and removed the reference to breaching as follows*

*line 165- 166: For the investigated events in this study, flood risk is mainly dominated by defence overflow.*

2) When describing the hydrodynamic model setting, at lines 168-169 I read "we did not represent the flow of water in the main channel. Rather boundary conditions were given as time series of water surface elevation imposed along the defence crests". These sentences are not clear at all. How was the upstream boundary condition (inflow discharge hydrographs from the hydrological model) used in hydrodynamic modelling? What is intended for "main channel"? Does the second sentence refer to downstream boundary condition only? (at lines 220-221 I read "simulated peak flow used as an upstream boundary condition in HEC-RAS" that states that the flow of water is somewhere represented).

*We thank the reviewer for this comment. We apologise for the confusion. Considering the reviewer comment, we removed the unclear sentence [previously line 168-169], and we rephrased the paragraph as follows*

*line 166- 169: The proposed analysis focussed upon the effects of extreme events that are so severe that all defences would, in any case, be overtopped. This allows for a simplification of the modelling problem and allows for a correct approximation of flows even without detailed bathymetric information in the main channel, as underlined in (Bates et al. 2005).*

3) The "Concluding remarks" has been enlarged, rather than improved. Now conclusions are long to read, contain repetitions and, finally, are unable to convey clear messages.

Each single paragraph is a collection of very different arguments, and concluding remarks on the same topic are dissected in different paragraphs. Please revise the structure of this last section.

*We thank the reviewer for this comment. We have reorganized and rewritten the "Concluding remarks" section, summarizing the main findings and trying to avoid repetitions.*

Minor Points
- L. 67: Abi-Samra, not Abi-Sarma.

*We fixed the reference.*

- L. 108: Skamarock, not Shamarock.

*We fixed the reference.*

- L. 128: Meehl et al., 2007, not 2017.

*We fixed the reference.*

- L. 166: Please consider adding a reference to Viero et al. (2019), as a relevant example of flooding dominated by defence overflow and defence breaching.

*We have rephrased this part of the text, and removed the references to defense breaching. Hence we did not add the reference*

- L. 167: Bates e al. (2013) is referenced in the text but, in the Bibliography, I can only find Bates et al. (2005). This item (line 480) is missing the title and is not properly formatted.

*We fixed the year in the text and modified the bibliography as well.*

- L. 258: check the reference to Figure 7 (maybe Figure 9 is the correct one).

*Thank you for the comment. We have fixed the figure number in the text.*

- L. 275: Xian et al. (2015), not (2005).

*We have fixed it.*

- L. 472: in the item Ahearn (2004) the report title is missing.

*We have removed the citation as we did not use this in the text.*

- l. 476: Barnard et al (2017) is not referenced in the text.

*We have removed the citation as we did not use this in the text.*

- l. 492: Bradbrook et al. (2004) is not referenced in the text. Moreover, the link provided is not the official one, please change it to https://doi.org/10.1080/15715124.2004.9635233.

*We have removed the reference.*

- L. 510: the year at the end.

*We fixed this. See line 472.*

- L. 514: Surname of the Authors first.

*We fixed this. See line 475.*

- L. 517: Danielson and Gesch is dated 2011, not 2016.

*We fixed this. See line 478.*

- L. 546: reference to Gerald et al. (2007), not cited in the text, is a duplicate of Meehl et al. (2007), and should be removed.

*We have fixed this and removed the duplicate reference.*

- L. 553-558: the reference to Hamman et al. (2016) is duplicated.

*We have removed the duplicate reference.*
- L. 605: O'Donnel (2020) is not referenced in the text.
*We have added the citation in the text. See line 204.*

- The two references to Schumann et al., 2007 should be denoted with 2007a and 2007b.

*We have fixed this as per reviewer's suggestion.*

- I found several references to a U.S.S Geological Survey throughout the Bibliography. It should read USGS, isn't it? (example l. 646)

*We fixed this.*

- L. 648-653: the reference to Vousdoukas et al. (2018) is duplicated.

*We removed the duplicate reference.*

- L. 657: Wahl et al. (2018) is not referenced in the text.

*We have removed it from bibliography.*

- L. 671-676: the reference to Ziervogel et al. (2014) is duplicated.

*We removed the duplicate reference.*

- Please note that in the Copernicus template there is a Bibliography style aimed at formatting the Bibliography with proper (and reader-friendly) indentation.

*Thank you for the suggestion. We have tried to follow the Bibliography style in the Copernicus template.*

- Figure 3 and throughout the text: put the superscript 3 in "m³/s".

*We have fixed it. Please see the next response.*

- Figure 3: "Stream flow at upstream" => "Upstream boundary condition"; "Total Water Level at Downstream" => "Downstream boundary condition"

*We have made changes according to reviewer's suggestions. Please see below-*

[Figure]

*Figure 3: Example of different scenarios showing the upstream boundary condition (top left-hand panel, including the discharge for actual Sandy and future Sandy), and downstream boundary (bottom left-hand panel, including tide, shifted tide, and shifted tide with SLR). Output flood extent is also shown (right-hand panel), including results for SD1 to SD5 [reader should refer to Tab. 3 and chapter 2.2 for specification on the scenarios]). Background map on the right-hand panel by ESRI web-services, provided by UConn/CTDEEP, Esri, Garmin, USGS, NGA, EPA, USDA, NPS*

- Caption of Figure 3: "firhg-hand panel" should read "right-hand panel".
*We fixed it. Please see the response above.*